# TIME-VARYING BAYESIAN OPTIMIZATION WITHOUT A METRONOME

## ABSTRACT

Time-Varying Bayesian Optimization (TVBO) is the go-to framework for optimizing a time-varying, expensive, noisy black-box function $f$. However, most of the asymptotic guarantees offered by TVBO algorithms rely on the assumption that observations are acquired at a constant frequency. As the GP inference complexity scales with the cube of its dataset size, this assumption is unrealistic in the long run. In this paper, we relax this assumption and derive the first upper regret bound that explicitly accounts for changes in the observations sampling frequency. Based on this analysis, we formulate practical recommendations about dataset sizes and stale data policies of TVBO algorithms. We illustrate how an algorithm (BOLT) that follows these recommendations performs better than the state-of-the-art of TVBO through experiments on synthetic and real-world problems.

## 1 INTRODUCTION

Many real-world problems require the optimization of a noisy, expensive-to-evaluate, black-box objective function $f : \mathcal{S} \subseteq \mathbb{R}^d \to \mathbb{R}$. When queried on an input $\boldsymbol{x} \in \mathcal{S}$, such a function only returns a noisy value $y(\boldsymbol{x}) = f(\boldsymbol{x}) + \epsilon$, where $\epsilon \sim \mathcal{N}(0, \sigma_0^2)$, but does not come with oracles that provide higher-order information such as $\nabla f(\boldsymbol{x})$ or $\nabla^2 f(\boldsymbol{x})$. In addition, observing $y(\boldsymbol{x})$ comes at a cost (time-wise and/or money-wise) that cannot be neglected. Examples of such problems are found in many areas, including robotics (Lizotte et al., 2007), computational biology (González et al., 2014), hyperparameters tuning (Bergstra et al., 2013) or computer networks (Si-Mohammed et al., 2024).

Bayesian Optimization (BO) is a black-box optimization framework that leverages a surrogate model of the objective function $f$ (usually a Gaussian Process (GP)) to simultaneously discover and optimize $f$. Because it has proven to be a powerful sample-efficient framework for optimizing black-boxes, BO is actually the go-to solution in a broad and diverse range of applications (Marchant & Ramos, 2012; Bardou et al., 2025; Wang et al., 2014). Its popularity and efficiency initiated a significant research effort dedicated to extend the BO framework to challenging contexts such as multi-objectives (Daulton et al., 2022) or high-dimensional input spaces (Bardou et al., 2024a). However, only a handful of papers study Time-Varying Bayesian Optimization (TVBO), where $f : \mathcal{S} \times \mathcal{T} \to \mathbb{R}$ is also a function of time in the temporal domain $\mathcal{T} \subseteq \mathbb{R}$. This is surprising given the ubiquity of time-varying black-box optimization problems in a variety of domains such as unmanned aerial vehicles (Melo et al., 2021), online clustering (Aggarwal et al., 2004) or network management (Kim et al., 2019).

Because of its temporal dimension, a TVBO task significantly differs from its static counterpart. In particular, recent works have provided evidence that *the response time $R(n)$*, which is a function of the dataset size $n$ and returns the time that separates two consecutive iterations, is a key feature of a TVBO algorithm. As $R(n)$ is typically in $\mathcal{O}(n^2)$,[1] finding a trade-off between a large $n$ and a small $R(n)$ is crucial for practical purposes, because it can yield significant performance gains (Bardou et al., 2024b). However, to the best of our knowledge, there is no analysis that relates $R(n)$ and the performance of TVBO algorithms unless $R(n) \in \Theta(1)$ is assumed. This paper fills this gap by:

- deriving the first (to the best of our knowledge) asymptotic regret bound that explicitly relates the performance of TVBO algorithms with their response times $R(n)$ (Section 3.2),

---

[1] Inverting the covariance matrix $k(\mathcal{D}, \mathcal{D})$ is actually $\mathcal{O}(n^3)$, but rank-one updates in the streaming setting make computing the precision matrix in $\mathcal{O}(n^2)$.

Table 1: Comparison of TVBO solutions. The solutions are listed in chronological order of publication and are compared on four different criteria: the assumptions they make on the objective function $f$ or on the response time of the algorithm, their handling of stale data and their regret guarantees. When applicable, the best value for each criterion appears **in bold**. UB stands for Upper Bound and a dagger † indicates that the result holds under highly restrictive assumptions.

| TVBO Solution | Restrictive Assump. on Objective $f$ | Assumption on Response Time | Stale Data Policy | Regret Guarantees |
|---|---|---|---|---|
| TV-GPUCB | Markov Model | $R(n) \in \Theta(1)$ | None | Linear UB |
| R-GPUCB | Markov Model | $R(n) \in \Theta(1)$ | Periodic Reset | Linear UB |
| ABO | **No** | **No** | None | None |
| SW-GPUCB | Finite Var. Budget | $R(n) \in \Theta(1)$ | Sliding Window | **Sublinear UB**$^\dagger$ |
| W-GPUCB | Finite Var. Budget | $R(n) \in \Theta(1)$ | None | **Sublinear UB**$^\dagger$ |
| ET-GPUCB | Markov Model | $R(n) \in \Theta(1)$ | Event-Based Reset | Linear UB |
| W-DBO | **No** | **No** | Relevancy-Based | None |
| BOLT | **No** | **No** | Relevancy-Based | Linear UB |

- exploiting these results to make recommendations for TVBO algorithms and embedding them into a new algorithm called BOLT (Sections 3.2, 3.3 and 4),
- providing evidence of the superiority of BOLT over the state-of-the-art of TVBO (Section 5).

## 2 BACKGROUND

### 2.1 TIME-VARYING BAYESIAN OPTIMIZATION

Like the vast majority of BO algorithms, a TVBO algorithm uses a GP as a surrogate model for the objective function $f$ with observational noise $\sigma_0^2$. A GP with mean 0 and covariance function $k$ (denoted by $\mathcal{GP}(0, k)$) is placed on $f$. Conditioned on a dataset of $n$ observations $\mathcal{D} = \{(\boldsymbol{x}_i, t_i, y_i)\}_{i \in [n]}$, where $y_i = f(\boldsymbol{x}_i, t_i) + \epsilon, \epsilon \sim \mathcal{N}(0, \sigma_0^2)$ for any $i \in [n]$, the posterior is $\mathcal{GP}(\mu_\mathcal{D}, \text{Cov}_\mathcal{D})$ where

$$\mu_\mathcal{D}(\boldsymbol{x}, t) = \boldsymbol{k}((\boldsymbol{x}, t), \mathcal{D})\boldsymbol{\Delta}^{-1}\boldsymbol{y}, \tag{1}$$

$$\text{Cov}_\mathcal{D}((\boldsymbol{x}, t), (\boldsymbol{x}', t')) = k((\boldsymbol{x}, t), (\boldsymbol{x}', t')) - \boldsymbol{k}^\top((\boldsymbol{x}, t), \mathcal{D})\boldsymbol{\Delta}^{-1}\boldsymbol{k}((\boldsymbol{x}', t'), \mathcal{D}), \tag{2}$$

where $\boldsymbol{\Delta} = \boldsymbol{K} + \sigma_0^2\boldsymbol{I}_n$, $\boldsymbol{K} = \boldsymbol{k}(\mathcal{D}, \mathcal{D})$, $\boldsymbol{k}(\mathcal{X}, \mathcal{Y}) = (k((\boldsymbol{x}_i, t_i), (\boldsymbol{x}_j, t_j)))_{(\boldsymbol{x}_i, t_i) \in \mathcal{X}, (\boldsymbol{x}_j, t_j) \in \mathcal{Y}}$, $\boldsymbol{y} = (y_i)_{y_i \in \mathcal{D}}$ and where $\boldsymbol{I}_n$ is the $n \times n$ identity matrix.

Equations (1) and (2) imply that, for any $(\boldsymbol{x}, t) \in \mathcal{S} \times \mathcal{T}$, $f(\boldsymbol{x}, t)|\mathcal{D} \sim \mathcal{N}(\mu_\mathcal{D}(\boldsymbol{x}, t), \sigma_\mathcal{D}^2(\boldsymbol{x}, t))$ where

$$\sigma_\mathcal{D}^2(\boldsymbol{x}, t) = \text{Cov}_\mathcal{D}((\boldsymbol{x}, t), (\boldsymbol{x}, t)). \tag{3}$$

At the $i$-th iteration occurring at time $t_i$, a TVBO algorithm looks for a query $(\boldsymbol{x}_i, t_i)$ that achieves the best exploration-exploitation trade-off. In the BO framework, this problem is addressed by maximizing an acquisition function $\alpha_\mathcal{D} : \mathcal{S} \times \mathcal{T} \to \mathbb{R}$, so that $\boldsymbol{x}_i = \arg\max_{\boldsymbol{x} \in \mathcal{S}} \alpha_\mathcal{D}(\boldsymbol{x}, t_i)$. Many acquisition functions have been proposed, the most popular being GPUCB (Srinivas et al., 2012), Expected Improvement (Mockus, 1994) and Probability of Improvement (Jones et al., 1998).

At the $i$-th iteration, the instantaneous performance of a TVBO algorithm is measured by the instantaneous regret $r_i = f(\boldsymbol{x}_i^*, t_i) - f(\boldsymbol{x}_i, t_i)$, where $\boldsymbol{x}_i^* = \arg\max_{\boldsymbol{x} \in \mathcal{S}} f(\boldsymbol{x}, t_i)$. The performance up to an horizon $T$ is usually measured by the cumulative regret $R_T = \sum_{i=1}^T r_i$.

### 2.2 STATE-OF-THE-ART OF TVBO

A synthetic comparison of all TVBO solutions in the literature is provided in Table 1. TVBO has been well studied in two settings: (i) under the assumption that the objective function $f$ follows a simple Markov model and (ii) under the assumption that $f$ has a finite variational budget. In setting (i), it

has been shown that any TVBO algorithm incurs a cumulative regret that is at least linear in the number of iterations, and three algorithms with a provable linear asymptotic regret bound have been proposed (Bogunovic et al., 2016; Brunzema et al., 2025). In setting (ii), sublinear regret bounds can be achieved (Zhou & Shroff, 2021; Deng et al., 2022). It should be noted that this setting is unrealistic in general, as it implies that the objective function becomes asymptotically static.

TVBO problems are characterized by three time scales: (i) *a temporal lengthscale* $l_T$ that can be likened to the inverse of the rate of change of $f$ in time, (ii) *the time horizon $H$* of the optimization task and (iii) *the response time of the algorithm $R(n)$* that aggregates the time needed for observing $f$ (i.e., $R(0)$) and the time needed for GP inference on a reasonable dataset size $n$. Let us discuss how the state-of-the-art of TVBO performs when these time scales vary.

**Near-Constant Response Time.** When $R(\lfloor H/R(0) \rfloor) \sim R(0)$, the cost of GP inference is negligible even when it is conducted on a dataset that contains the maximal number of observations that can be collected within the time horizon $H$ (that is, $\lfloor H/R(0) \rfloor$). This happens when the objective function $f$ is very expensive to evaluate. In this setting, it is reasonable to assume that the response time $R(n)$ is constant and to expect that all algorithms perform well. Note that all TVBO algorithms that provide a regret bound assume $R(n) \in \Theta(1)$ (see Table 1).

**Quasi-Static Problems.** When $l_T \gg R(n)$ for a reasonable dataset size $n \ll H$, observations can be acquired rapidly with respect to the rate of change of the objective function. Although the time taken by GP inference may not be negligible (i.e., $R(n) \in \Theta(1)$ may no longer be assumed), TVBO algorithms that address the time-varying problem as a series of static optimization problems coupled with periodic resets of their datasets (e.g., R-GPUCB, ET-GPUCB) may still perform well, even though they remove relevant observations at each reset. However, TVBO algorithms that never remove observations from their datasets (e.g., TV-GPUCB, ABO) may eventually become prohibitive to use.

**All Other Scenarios.** In all other scenarios (e.g., large time horizon $H$, $l_T \sim R(n)$ for a reasonable dataset size $n$), most TVBO algorithms in the state-of-the-art are expected to be suboptimal since the response time cannot safely be assumed constant (TV-GPUCB and ABO may eventually become prohibitive) and $f$ cannot be reasonably seen as quasi-static (R-GPUCB and ET-GPUCB may not be able to learn anything meaningful before triggering another reset of their datasets). W-DBO may still perform well in these settings, but its good empirical performance has not been explained formally.

In this paper, we conduct the first regret analysis that is explicitly built on the response time of TVBO algorithms, without assuming a finite variational budget for $f$ or a constant response time $R(n)$. This analysis allows us to make recommendations for TVBO solutions regarding their maximal dataset sizes (Section 3.2) and their policies to deal with stale observations (Section 3.3). Finally, we design BOLT, a TVBO algorithm that follows our recommendations (Section 4) and demonstrate its competitiveness against the state-of-the-art of TVBO on quasi-static, near-constant response time and other types of synthetic and real-world problems (Section 5).

## 3 MAIN RESULTS

### 3.1 CORE ASSUMPTIONS

In this section, we introduce the core assumptions underpinning this work. Assumption 3.1 is made in all GP-based BO papers as it ensures the existence of the surrogate model. Assumption 3.2 is widely used in the TVBO literature (Bogunovic et al., 2016; Nyikosa et al., 2018; Bardou et al., 2024b). It captures spatio-temporal dynamics with two correlation functions $k_S$ and $k_T$. Assumption 3.3 requires that $k_S$ and $k_T$ verify some properties. It holds for most kernels used in practice (e.g., Matérn, RBF, Rational Quadratic). Assumption 3.4 introduces some regularity about the objective function in the spatial domain $\mathcal{S}$. It is used for most regret proofs in the literature (Srinivas et al., 2012; Bogunovic et al., 2016) and is satisfied by sufficiently smooth GPs (e.g., built with Matérn with $\nu > 2$, RBF) but it can fail with rougher GPs (e.g., Ornstein-Uhlenbeck processes built with Matérn kernels where $\nu = 1/2$).

**Assumption 3.1** (Surrogate Model). The objective function $f : \mathcal{S} \times \mathcal{T} \to \mathbb{R}$ is a $\mathcal{GP}(\mu_0, k)$, where $\mathcal{S} = [0,1]^d$, $\mu_0 = 0$ without loss of generality and $k : (\mathcal{S} \times \mathcal{T})^2 \to \mathbb{R}$ is a covariance function.

**Assumption 3.2** (Covariance Separability). The covariance function $k$ has the following form:

$$k((\boldsymbol{x}, t), (\boldsymbol{x}', t')) = \lambda k_S(\boldsymbol{x}, \boldsymbol{x}') k_T(t, t')$$

where, without loss of generality, $\lambda = 1$ is the signal variance while $k_S : \mathcal{S} \times \mathcal{S} \to [-1, 1]$ and $k_T : \mathcal{T} \times \mathcal{T} \to [-1, 1]$ are correlation functions in the spatial and temporal domain, respectively.

**Assumption 3.3** (Covariance Properties). The correlation function $k_S$ (resp., $k_T$) is real, even, stationary and can therefore be rewritten as $k_S(\boldsymbol{x}, \boldsymbol{x}') = k_S(\boldsymbol{x} - \boldsymbol{x}')$ for all $\boldsymbol{x}, \boldsymbol{x}' \in \mathcal{S}$ (resp., $k_T(t, t') = k_T(|t - t'|)$ for all $t, t' \in \mathcal{T}$). Furthermore, $\min_{\boldsymbol{x}, \boldsymbol{x}' \in \mathcal{S}} k_S(\boldsymbol{x}, \boldsymbol{x}') > 0$, $k_T$ is isotropic and $\mathrm{supp}(S_T) \subseteq \mathbb{R}$ is a (potentially unbounded) interval, where $S_T$ is the Fourier transform of $k_T$.

**Assumption 3.4** (Lipschitz Continuity in $\mathcal{S}$). Let $g \sim \mathcal{GP}(0, k_S)$. Then, for any $\boldsymbol{x} \in \mathcal{S}$, any $L > 0$ and any $i \in [d]$,

$$\mathbb{P}\left[\left|\frac{\partial g(\boldsymbol{x})}{\partial x_i}\right| > L\right] \leq a e^{-(L/b)^2}.$$

Finally, let us properly define the response time $R$ of a TVBO algorithm, which satisfies three natural properties: (i) the objective function cannot be observed at an arbitrarily high frequency because querying it takes a nonzero amount of time, (ii) the response time is an increasing function of the dataset size $n$ and (iii) when the dataset size $n$ diverges, the response time diverges as well.

**Definition 3.5** (Response Time). The response time $R : \mathbb{N} \to \mathbb{R}^+$ is a function that returns the time separating two consecutive iterations of a TVBO algorithm with a dataset of size $n \in \mathbb{N}$ such that (i) $R(0) > 0$, (ii) $\forall n \in \mathbb{N}, R(n+1) \geq R(n)$ and (iii) $\lim_{n \to \infty} R(n) = +\infty$.

### 3.2 Regret Analysis Without a Metronome

We now analyze the regret of a TVBO algorithm under the assumptions and definitions introduced in Section 3.1. As the cumulative regret of a TVBO algorithm is linear in the worst case when $R(n) \in \Theta(1)$ Bogunovic et al. (2016), one can expect a similar result when $R(n) \in \omega(1)$.

To be able to track the maximal argument of $f$ in the long run, a TVBO algorithm must prevent its dataset size from diverging. As reported by Table 1, the TVBO literature proposes two policies to achieve that: (i) reset the dataset periodically (Bogunovic et al., 2016) or after an event is triggered (Brunzema et al., 2025) and (ii) delete observations on the fly based on a removal budget (Bardou et al., 2024b). Under the assumption that $k_T$ is an exponential kernel and $R(n) \in \Theta(1)$, Bogunovic et al. (2016) and Brunzema et al. (2025) showed that the policy (i) incurs a linear cumulative regret. As the latest empirical evidence suggests that the best policy is to remove observations on the fly (Bardou et al., 2024b), we derive an asymptotic regret bound for any TVBO algorithm that adopts policy (ii) to make sure that its dataset size does not exceed a maximal size.

**Theorem 3.6.** *Let $\mathcal{A}$ be a TVBO algorithm that uses the GPUCB acquisition function. Let $R(s)$ be the response time of $\mathcal{A}$ for a dataset size $s \in \mathbb{N}$. Let $n$ be the maximal dataset size of $\mathcal{A}$ and let $\|\boldsymbol{u}_n\|_2^2 = \sum_{i=1}^n k_T^2(iR(n))$. Pick $\delta \in (0, 1)$ and let $R_T$ be the cumulative regret of $\mathcal{A}$ after $T$ iterations. Then, with probability $1 - \delta$,*

$$R_T \leq \sqrt{TC_1\beta_T\left(\gamma_n + \frac{\sigma_0^{-2}}{2}(T - n)(1 - C_2\|\boldsymbol{u}_n\|_2^2)\right)} + \frac{\pi^2}{6} \tag{4}$$

*where $C_1, C_2 \in \mathcal{O}(1)$, $\gamma_n \in \mathcal{O}(n)$ and $\beta_T \in \mathcal{O}(\log T)$ are defined in Appendix A.*

The proof of Theorem 3.6 is provided in Appendix A. In essence, we start by bounding the cumulative regret of the TVBO algorithm from above with an expression that involves the mutual information $\gamma_n = I(\boldsymbol{f}_n, \boldsymbol{y}_n)$, where $\boldsymbol{f}_n = (f(\boldsymbol{x}_1, t_1), \cdots, f(\boldsymbol{x}_n, t_n))$ and $\boldsymbol{y}_n = (y_1, \cdots, y_n)$, and the posterior variance of the surrogate model with maximal dataset size $n$. This is achieved with proof techniques introduced by Srinivas et al. (2012); Bogunovic et al. (2016). Then, we derive an upper bound for the posterior variance that involves $\|\boldsymbol{u}_n\|_2^2$. Combining these two upper bounds yields (4).

Two things are interesting to note about Theorem 3.6. First, although potentially loose, the provided bound is, to the best of our knowledge, the first connection between the regret, the temporal correlation structure described by the temporal kernel $k_T$ and the response time of the TVBO algorithm. Second, it recovers the well-known sublinear growth for the cumulative regret of GPUCB in the static case.

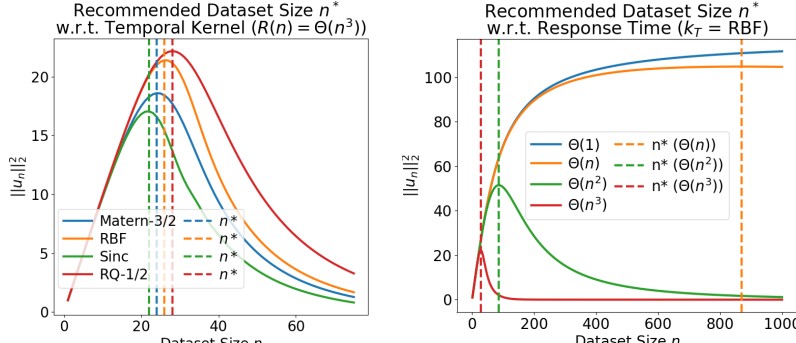

Figure 1: Recommended dataset size $n^* = \arg\max_{n \in \mathbb{N}} ||\boldsymbol{u}_n||_2^2$. (Left) Recommended dataset size for several common temporal covariance functions $k_T$, under the assumption that the response time is $R(n) \in \Theta(n^3)$. (Right) The recommended dataset size for several response times, under the assumption that $k_T$ is an RBF covariance function.

In fact, when $k_T(t, t') = 1$ for any $t, t' \in \mathcal{T}$, $\gamma_n$ is the mutual information associated with $k_S$ and setting $n = T$ (removing the constraint of a maximal dataset size) yields $R_T \leq \sqrt{TC_1 \beta_T \gamma_T} + \pi^2/6$, which is exactly the upper bound of $R_T$ obtained in Theorem 2 of Srinivas et al. (2012).

For a fixed dataset size $n$ and in the long-term regime (i.e., when $T \gg n$), $\gamma_n$ is a dominated constant and $T - n \sim T$. Therefore, Theorem 3.6 immediately yields $R_T \in \mathcal{O}\left(T\left(1 - C_2||\boldsymbol{u}_n||_2^2\right)\right)$ (we neglect constant and polylogarithmic terms for the sake of simplicity). This linear upper regret bound is in line with the other regret bounds derived in the TVBO literature when the response time $R(n)$ is constant with respect to $n$ (Bogunovic et al., 2016; Brunzema et al., 2025). As a byproduct of Theorem 3.6, we also deduce a recommendation for the maximal dataset size of a TVBO algorithm with temporal kernel $k_T$ and response time $R(n)$. In fact, there is a value of $n$ that minimizes (4), which leads to the following recommendation.

**Recommendation 1.** A TVBO algorithm should maintain a dataset of size $n^*$, where

$$n^* = \arg\max_{n \in \mathbb{N}} ||\boldsymbol{u}_n||_2^2 = \arg\max_{n \in \mathbb{N}} \sum_{i=1}^{n} k_T^2(iR(n)). \tag{5}$$

Finding a closed form for $n^*$ is difficult. However, it can be easily shown that the relaxed form of the function to maximize in (5), that is, $g(x) = \int_1^x k_T^2(sR(x))ds$, has a unique maximum $x^*$ under Assumption 3.3 and Definition 3.5. Consequently, (5) can be found using direct search and the finite difference $\Delta u(n) = ||\boldsymbol{u}_{n+1}||_2^2 - ||\boldsymbol{u}_n||_2^2$. Starting from an initial guess $n_0 \in \mathbb{N}$, the sequence formed by iteratively applying $n_{i+1} = n_i + \text{sign}(\Delta u(n))$ will converge to $n^*$. For the sake of intuition, we provide a short sketch of proof of this fact when $k_T$ is nonnegative and monotonically decreasing.

**Convergence of Direct Search.** Let us consider the relaxed function to maximize in (5), that is $g(x) = \int_0^x k_T^2(sR(x))ds$ and let us study its derivative. An application of Leibniz rule yields

$$g'(x) = \underbrace{k_T^2(xR(x))}_{\geq 0} + \underbrace{2R'(x)}_{>0} \int_0^x \underbrace{k_T(sR(x))s}_{\geq 0} \underbrace{k_T'(sR(x))}_{\leq 0} ds. \tag{6}$$

A simple sign analysis in (6) coupled with the fact that $g''(x)$ is non-positive yields that if $g(x)$ has a maximum (which occurs when $R(x) \in \omega(1)$), then it is unique and therefore, global. This immediately leads to the discretized problem (5) to have either (i) a unique maximum (which is then global) whose argument is $n^*$ or (ii) two maxima of equal values, the arguments of which are consecutive integers (i.e., $n^*$ and $n^* + 1$). Because the direct search procedure we propose converges to a maximum, it converges to $n^*$.

Figure 1 illustrates how $||\boldsymbol{u}_n||_2^2$ and $n^*$ behave under different covariance functions and response times. The left panel shows that the smoothest temporal covariance functions (e.g., RBF or Rational

Quadratic) also yield the largest $n^*$. This is intuitive because a smoother GP surrogate can extract useful information from observations collected further in the past. The right panel shows that the response time of the TVBO algorithm has a significant impact on $||\boldsymbol{u}_n||_2^2$ and $n^*$: response times that scale slowly with the dataset size $n$ lead to larger $n^*$. In the case where $R(n)$ does not scale with the dataset size $n$ (i.e., a constant response time $R(n) \in \Theta(1)$), $||\boldsymbol{u}_n||_2^2$ is always increasing; therefore, an infinite dataset size is recommended.

### 3.3 REMOVAL OF IRRELEVANT OBSERVATIONS

In the previous section, we have made a recommendation about the dataset size of a TVBO algorithm, but we have not provided a policy to select which observations to remove from the dataset $\mathcal{D}$. The intuitive policy would be to remove the oldest observations in the dataset, but recent works provide empirical evidence that this is suboptimal.

Instead, Bardou et al. (2024b) propose to remove the observation $\boldsymbol{o}_i \in \mathcal{D}$ that minimizes the integrated 2-Wasserstein distance (Kantorovich, 1960) between two GP surrogates: $\mathcal{GP}_{\mathcal{D}} \left( \mu_{\mathcal{D}}, \sigma_{\mathcal{D}}^2 \right)$ conditioned on the dataset $\mathcal{D}$ and $\mathcal{GP}_{\tilde{\mathcal{D}}} \left( \mu_{\tilde{\mathcal{D}}}, \sigma_{\tilde{\mathcal{D}}}^2 \right)$ conditioned on $\tilde{\mathcal{D}} = \mathcal{D} \setminus \{\boldsymbol{o}_i\}$. However, they do not justify why removing these observations is effective. In this section, we provide a rigorous justification for this policy, under the following assumption.

**Assumption 3.7** (Lipschitz Acquisition). At iteration $T \in \mathbb{N}$, the acquisition function $\alpha : \mathcal{S} \times \mathcal{T} \to \mathbb{R}$ is built from the posterior mean $\mu(\boldsymbol{x}, t)$ and the posterior standard deviation $\sigma(\boldsymbol{x}, t)$ of the GP surrogate, that is, $\alpha(\boldsymbol{x}, t) = g_T(\mu(\boldsymbol{x}, t), \sigma(\boldsymbol{x}, t))$ for some function $g_T : \mathbb{R}^2 \to \mathbb{R}$. Furthermore, $g_T$ is a Lipschitz continuous function, that is,

$$|g_T(\boldsymbol{u}) - g_T(\boldsymbol{v})| \le L_T ||\boldsymbol{u} - \boldsymbol{v}||_2, \ \forall \boldsymbol{u}, \boldsymbol{v} \in \mathbb{R}^2. \tag{7}$$

Assumption 3.7 holds for many acquisition functions. For example, if $\alpha$ is GPUCB (Srinivas et al., 2012), that is, $\alpha(\boldsymbol{x}, t) = \mu(\boldsymbol{x}, t) + \beta_T^{1/2} \sigma(\boldsymbol{x}, t)$, then Assumption 3.7 holds with $L_T = \sqrt{1 + \beta_T}$.

**Theorem 3.8.** *Let $\alpha$ be an acquisition function satisfying Assumption 3.7. Let us denote by $\alpha_{\mathcal{D}}$ (resp., $\alpha_{\tilde{\mathcal{D}}}$) the acquisition function $\alpha$ exploiting the surrogates $\mathcal{GP}_{\mathcal{D}}$ (resp., $\mathcal{GP}_{\tilde{\mathcal{D}}}$). Then, on any subset $\mathcal{S}' \times \mathcal{T}'$ of the spatio-temporal domain $\mathcal{S} \times \mathcal{T}$:*

$$||\alpha_{\mathcal{D}} - \alpha_{\tilde{\mathcal{D}}}||_2 \le L_T W_2 \left( \mathcal{GP}_{\mathcal{D}}, \mathcal{GP}_{\tilde{\mathcal{D}}} \right), \tag{8}$$

*where $||\alpha_{\mathcal{D}} - \alpha_{\tilde{\mathcal{D}}}||_2$ is the $L^2$ distance between $\alpha_{\mathcal{D}}$ and $\alpha_{\tilde{\mathcal{D}}}$ on $\mathcal{S}' \times \mathcal{T}'$ and $W_2 \left( \mathcal{GP}_{\mathcal{D}}, \mathcal{GP}_{\tilde{\mathcal{D}}} \right)$ is the integrated 2-Wasserstein distance between $\mathcal{GP}_{\mathcal{D}}$ and $\mathcal{GP}_{\tilde{\mathcal{D}}}$ on the same domain $\mathcal{S}' \times \mathcal{T}'$.*

Theorem 3.8 is proven in Appendix B. Assumption 3.7 allows to upper bound the $L^2$ distance between $\alpha_{\mathcal{D}}$ and $\alpha_{\tilde{\mathcal{D}}}$ on any subset of $\mathcal{S} \times \mathcal{T}$ in terms of the posterior means and variances of $\mathcal{GP}_{\mathcal{D}}$ and $\mathcal{GP}_{\tilde{\mathcal{D}}}$, which in turn naturally leads to the integrated Wasserstein distance between the GP surrogates.

Theorem 3.8 provides an interesting connection between the discrepancy of the acquisition functions built from two surrogate models $\mathcal{GP}_{\mathcal{D}}$ and $\mathcal{GP}_{\tilde{\mathcal{D}}}$, and the integrated 2-Wasserstein distance between these two surrogates. Although potentially loose, this connection is a justification for the following stale data management policy: given a maximal dataset size $n$ and a dataset $\mathcal{D}$ of size $T$ (where $T > n$), one should build $\tilde{\mathcal{D}} = \mathcal{D} \setminus X^*$, where $X^* = \arg\min_{X \in 2^{\mathcal{D}}, |X|=T-n} W_2(\mathcal{GP}_{\mathcal{D}}, \mathcal{GP}_{\mathcal{D} \setminus X})$. Unfortunately, and similarly to GP inference with an ever-growing dataset, this policy would become intractable in the long run. Therefore, for dealing with stale data, we recommend the same greedy heuristic as the one introduced in Bardou et al. (2024b).

**Recommendation 2.** Let $\tilde{\mathcal{D}}_i$ be the dataset where one observation $\boldsymbol{o}_i$ has been removed from the dataset $\mathcal{D} = \{\boldsymbol{o}_1, \cdots, \boldsymbol{o}_n\}$. Then, choosing to remove $\boldsymbol{o}_{i^*}$, where $i^* = \arg\min_{i \in [n]} W_2 \left( \mathcal{GP}_{\mathcal{D}}, \mathcal{GP}_{\tilde{\mathcal{D}}_i} \right)$, minimizes the effect of the removal on the acquisition function $\alpha$. Note that $i^*$ can be found by an online algorithm, as in Bardou et al. (2024b).

## 4 BOLT: BAYESIAN OPTIMIZATION IN THE LONG TERM

In the previous section, we have made two recommendations about the dataset size and the observation removal policy for TVBO algorithms, which are only useful when the optimization task is carried out

---

**Algorithm 1** BOLT

    **Input:** objective $f : \mathcal{S} \times \mathcal{T} \to \mathbb{R}$, acquisition function $\alpha$, clock $\mathcal{C}$
    Init dataset $\mathcal{D} = \emptyset$
    **while true do**
       Get current time $t$ from $\mathcal{C}$
       Find $x_t = \arg\max_{\boldsymbol{x} \in \mathcal{S}} \alpha_{\mathcal{D}}(\boldsymbol{x}, t)$
       Observe $y = f(\boldsymbol{x}_t, t) + \epsilon$
       $\mathcal{D} = \mathcal{D} \cup \{((\boldsymbol{x}_t, t), y)\}$
       Infer GP hyperparameters and response time $R(n)$ from data
       Find the recommended maximum dataset size $n^*$ with (5)
       **if** $|\mathcal{D}| > n^*$ **then**
          Find $\boldsymbol{o}^* = \arg\min_{\boldsymbol{o} \in \mathcal{D}} W_2 \left( \mathcal{GP}_{\mathcal{D}}, \mathcal{GP}_{\mathcal{D} \setminus \{\boldsymbol{o}\}} \right)$
          $\mathcal{D} = \mathcal{D} \setminus \{\boldsymbol{o}^*\}$
       **end if**
    **end while**

---

in the long term, that is, over a sufficiently long period of time so that removing observations from $\mathcal{D}$ produces a noticeable effect on the response time $R(n)$ and, therefore, on performance.

In this section, we propose BOLT, a new TVBO algorithm that follows these two recommendations. The pseudocode is provided in Algorithm 1. At each iteration, BOLT follows three simple steps: (i) at time $t$, find a promising query $(\boldsymbol{x}_t, t)$, (ii) observe the response $f(\boldsymbol{x}_t, t)$ and augment $\mathcal{D}$ with the collected observation and (iii) remove stale observations if necessary. Step (iii) unfolds as follows. First, use Recommendation 1 to infer a maximal dataset size based on the temporal correlation structure $k_T$ and the response time $R(n)$. Second, use Recommendation 2 to enforce this maximal dataset size on $\mathcal{D}$ by removing the most stale observations.

The algorithm is based on W-DBO (Bardou et al., 2024b), but replaces the arbitrary budget used by W-DBO to remove stale observations by the principled maximal dataset size deduced from Theorem 3.6. This has important implications in practice. For example, unlike W-DBO, BOLT can be used on widely different devices as the maximal dataset size (5) naturally adapts to different response times (which may be affected by the computing powers of the devices).

**Estimating the Reponse Time.** In Algorithm 1, $R(n)$ is inferred in real-time, similarly to the GP surrogate hyperparameters (e.g., the variance $\lambda$, the spatial and temporal lengthscales for $k_S$ and $k_T$, the noise level $\sigma_0^2$). At the beginning of any iteration on dataset $\mathcal{D}$ (and dataset size $|\mathcal{D}|$), the corresponding response time $R(|\mathcal{D}|)$ is measured as the duration between two consecutive calls to the clock $\mathcal{C}$ used in Algorithm 1 to obtain the current time. Next, the observation $(|\mathcal{D}|, R(|\mathcal{D}|))$ is added to another dataset, denoted by $\mathcal{R}$, which is fed to a regression model that infers the response time of BOLT. Remember that the computational complexity of GP inference scales at most as $\mathcal{O}(|\mathcal{D}|^3)$. Therefore, we conduct a 3rd-degree polynomial regression on the dataset $\mathcal{R}$ to estimate the response time $R(|\mathcal{D}|)$ used in BOLT.

## 5 NUMERICAL RESULTS

In this section, we evaluate BOLT against the state-of-the-art of TVBO. All algorithms in the TVBO literature that use an infinite variational budget for the objective function $f$ are considered, namely ABO (Nyikosa et al., 2018), ET-GPUCB (Brunzema et al., 2025), W-DBO (Bardou et al., 2024b), TV-GPUCB and R-GPUCB (Bogunovic et al., 2016). As a control solution, we also include the vanilla GPUCB algorithm (Srinivas et al., 2012).

### 5.1 EXPERIMENTAL SETTING

We evaluate the TVBO algorithms on a set of 10 synthetic and real-world benchmarks, which are thoroughly described in Appendix C. Some can be viewed as either quasi-static or as having a near-constant reponse time (see Section 2 for definitions of these types of problems). Each benchmark

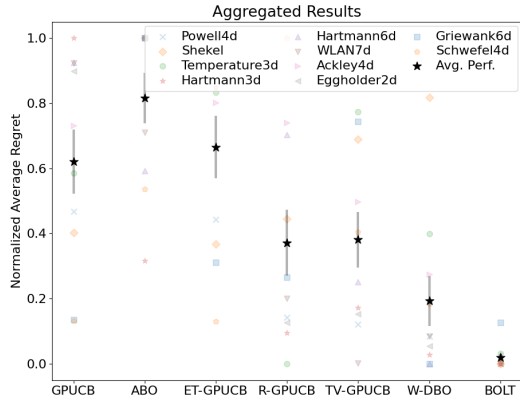

Figure 2: Normalized average regret across the benchmarks (lower is better). For each benchmark, the best performing TVBO algorithm gets a normalized regret of 0, and the worst performing TVBO algorithm gets a normalized regret of 1. The normalized regrets are then averaged across all the benchmarks.

is a $(d+1)$-dimensional function to optimize. The first $d$ dimensions make up the spatial domain $\mathcal{S}$, which is normalized in $[0, 1]^d$. The $(d+1)$th dimension is the time dimension.

Each experiment begins with a warm-up phase consisting of 15 random queries. Then, at each iteration, the TVBO algorithms must perform the following tasks in real-time:

- **Hyperparameters inference**: the variance $\lambda$, the spatial lengthscale $l_S$ and the observational noise $\sigma_0^2$ must always be inferred. When applicable, the temporal lengthscale $l_T$ and the response time $R(n)$ of the algorithm are also inferred at this stage. Unless requested otherwise by the algorithm, each solution uses a Matérn-5/2 spatial covariance function, and a Matérn-3/2 temporal covariance function.

- **Optimization of the acquisition function**: the acquisition function is always GPUCB (Srinivas et al., 2012), and the next observation $\boldsymbol{x}_t$ is found by using multi-start gradient ascent.

- **Function observation**: a noisy function value $y_i = f(\boldsymbol{x}_i, t_i) + \epsilon$ is observed, and the observation $(\boldsymbol{x}_i, t_i, y_i)$ is added to the dataset $\mathcal{D}$.

- **Dataset cleaning**: when applicable, this is the stage in which some observations in $\mathcal{D}$ might be removed.

Each TVBO algorithm has been implemented with the same BO framework, namely BOTorch (Balandat et al., 2020). Moreover, for the sake of a fair evaluation, the critical, time-consuming stages (i.e., hyperparameters inference and acquisition function optimization) are performed using the same BOTorch routines. Finally, all experiments have been independently replicated 10 times on a laptop equipped with an Intel Core i9-9980HK @ 2.40 GHz with 8 cores (16 threads). No graphics card was used to speed up GP inference.

## 5.2 EXPERIMENTAL RESULTS

The average regrets (and their standard errors) for each TVBO algorithm on each benchmark are listed in Table 2. BOLT performs consistently well: it is either the best or second-best performing TVBO algorithm on each benchmark, leading to a high average performance across all benchmarks, as reported in the last row of Table 2 and visualized in Figure 2.

This is largely explained by the fact that BOLT is able to adapt to the landscape of the objective function. As an example, Figure 3 shows the evolution of the dataset size of each TVBO algorithm for the Eggholder and Powell synthetic functions.

When optimizing Eggholder (left plot of Figure 3), BOLT removes a significant number of observations. This is expected because Eggholder is an erratic function with multiple local optima, thus an observation rapidly becomes irrelevant to predict the future behavior of the objective function. This

Table 2: Comparison of BOLT against state-of-the-art TVBO algorithms. For each experiment and each algorithm, the average regret and its standard error over 10 independent experiments is provided (lower is better). For each experiment, the performance of the best algorithm is written **in bold**, and the performance of algorithms whose confidence intervals overlap the confidence interval of the best performing algorithm are underlined.

| BENCHMARK ($d+1$) | GPUCB | ET-GPUCB | R-GPUCB | TV-GPUCB | ABO | W-DBO | BOLT |
|---|---|---|---|---|---|---|---|
| SHEKEL (4) | 2.57 ±0.05 | 2.57 ±0.04 | 2.58 ±0.02 | 2.62 ±0.05 | 2.67 ±0.11 | 2.64 ±0.04 | **2.51** **±0.02** |
| HARTMANN (3) | 1.58 ±0.08 | 1.43 ±0.12 | 0.70 ±0.05 | 0.77 ±0.10 | 0.91 ±0.12 | 0.63 ±0.04 | **0.60** **±0.04** |
| ACKLEY (4) | 4.26 ±0.50 | 4.39 ±0.45 | 4.28 ±0.20 | 3.83 ±0.57 | 4.75 ±0.53 | 3.42 ±0.46 | **2.92** **±0.35** |
| GRIEWANK (6) | 0.59 ±0.01 | 0.61 ±0.01 | 0.61 ±0.01 | 0.66 ±0.02 | 0.69 ±0.03 | **0.57** **±0.03** | 0.58 ±0.02 |
| EGGHOLDER (2) | 517 ±16.4 | 522 ±13.2 | 298 ±4.3 | 305 ±13.3 | 546 ±109.9 | 277 ±12.4 | **262** **±7.1** |
| SCHWEFEL (4) | 590 ±20.9 | 589 ±35.3 | 1024 ±10.3 | 726 ±34.1 | 792 ±61.3 | 615 ±21.5 | **523** **±30.3** |
| HARTMANN (6) | 1.67 ±0.01 | 1.75 ±0.07 | 1.44 ±0.02 | 0.96 ±0.24 | 1.32 ±0.17 | **0.70** **±0.03** | 0.72 ±0.06 |
| POWELL (4) | 3429 ±153 | 3283 ±240 | 1422 ±64 | 1301 ±155 | 6721 ±2113 | 1066 ±149 | **547** **±166** |
| TEMPERATURE (3) | 1.12 ±0.05 | 1.29 ±0.11 | **0.70** **±0.01** | 1.25 ±0.10 | 1.41 ±0.26 | 0.98 ±0.06 | 0.72 ±0.03 |
| WLAN (7) | 20.2 ±1.58 | 21.2 ±1.05 | 10.4 ±0.16 | 7.8 ±0.34 | 17.3 ±1.93 | 8.9 ±0.20 | **7.7** **±0.15** |
| **OVERALL** | 0.62 ±0.10 | 0.67 ±0.10 | 0.37 ±0.10 | 0.38 ±0.08 | 0.82 ±0.08 | 0.19 ±0.08 | **0.02** **±0.01** |

gives a significant advantage to TVBO algorithms that are able to remove observations on the fly (R-GPUCB, W-DBO, BOLT), as illustrated by the corresponding results in Table 2.

In contrast, when optimizing Powell (right plot of Figure 3), BOLT is able to adapt its policy and behaves similarly to algorithms that never remove an observation from their datasets (such as TV-GPUCB). This is also expected behavior, because Powell is much smoother than Eggholder, hence an observation remains relevant to predict the future behavior of the objective function even after a long period of time. The corresponding results reported in Table 2 indicate that BOLT performs significantly better than TV-GPUCB, although their dataset sizes are roughly similar. This can be explained by the superior quality of the surrogate model of BOLT, which is governed by the mild Assumptions 3.1, 3.2 and 3.3 only. In contrast, GPUCB does not capture temporal dynamics, and TV-GPUCB does so under more restrictive assumptions (Markovian setting (Bogunovic et al., 2016)).

Finally, some of our benchmarks can be viewed as quasi-static (e.g., Temperature) or as having a near-constant response time (e.g., Shekel), as defined in Section 2 and reported in Table 3 from Appendix C. As expected, the performance of all TVBO algorithms is pretty similar for near-constant problems (except for ET-GPUCB that encounters problems with on-the-fly hyperparameter inference as discussed by the authors in Brunzema et al. (2025)). On quasi-static problems, TVBO algorithms that do remove observations (R-GP-UCB, W-DBO, BOLT) are the best-performing ones. Table 2 clearly shows that, regardless of the nature of the problem (quasi-static, near-constant, miscellaneous), BOLT is the only algorithm able to consistently achieve competitive or superior performance.

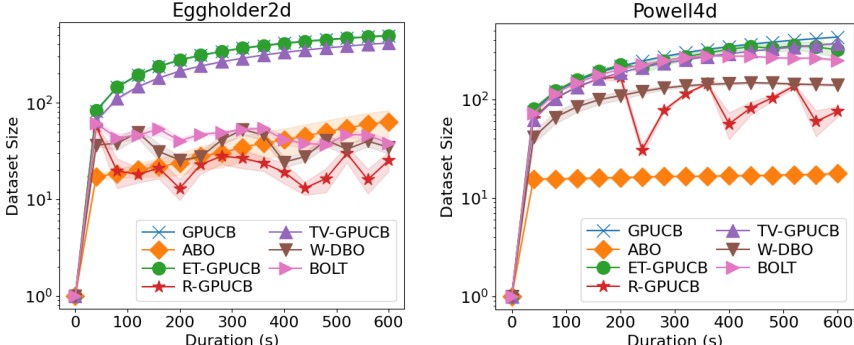

Figure 3: Evolution of the dataset sizes $n$ of the TVBO algorithms on the Eggholder (left) and Powell (right) synthetic functions. The plots are in log scale.

## 6 CONCLUSION

In this paper, we proposed a regret analysis that, unlike most existing results in the literature, explicitly incorporates the effect of the sampling frequency in TVBO, which naturally decreases as the dataset grows. Our main contribution, Theorem 3.6, establishes an asymptotic upper bound on the regret of any TVBO algorithm that controls its dataset size by removing observations on the fly. This result aligns with existing bounds in the static BO literature (Srinivas et al., 2012) and the TVBO literature (Bogunovic et al., 2016), while also enabling principled recommendations for observation-removal policies in TVBO. To demonstrate its practical impact, we introduced BOLT, a TVBO algorithm designed according to these recommendations, and showed that it consistently outperforms the state of the art on a diverse set of synthetic and real-world benchmarks.

In future work, we plan to study the impact of sampling frequency on TVBO algorithms that relax Assumptions 3.2 and 3.3. As an example, considering a non-separable spatio-temporal covariance function seems to be a particularly interesting research avenue since it allows the TVBO algorithm to encode more complex spatio-temporal dynamics. Also, studying how the cumulative regret of TVBO algorithms behaves with respect to the sampling frequency of observations is a promising research avenue that is likely to improve our understanding of TVBO.

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

# A    UPPER REGRET BOUND WITHOUT A METRONOME

In this appendix, we provide all the details required to prove Theorem 3.6. For the sake of completeness, we start by deriving the usual instantaneous regret bound provided in most regret proofs that involve GPUCB (Srinivas et al., 2012; Bogunovic et al., 2016).

**Lemma A.1.** *Let $r_n = f(\boldsymbol{x}_n^*, t_n) - f(\boldsymbol{x}_n, t_n)$ be the instantaneous regret at the $n$-th iteration of the TVBO algorithm $\mathcal{A}$, where $\boldsymbol{x}_n^* = \arg\max_{\boldsymbol{x} \in \mathcal{S}} f(\boldsymbol{x}, t_n)$, $\boldsymbol{x}_n = \arg\max_{\boldsymbol{x} \in \mathcal{S}} \alpha_{\mathcal{D}}(\boldsymbol{x}, t_n)$, $\alpha_{\mathcal{D}}$ is GPUCB computed on $\mathcal{D}$ and where $\mathcal{D}$ is a dataset of observations collected with the distribution $\mu$. Pick $\delta \in (0, 1)$, then with probability at least $1 - \delta$,*

$$r_n \le 2\beta_n^{1/2}\sigma_{\mathcal{D}}(\boldsymbol{x}_n, t_n) + \frac{1}{n^2} \tag{9}$$

*where $\sigma_{\mathcal{D}}(\boldsymbol{x}, t)$ is the posterior standard deviation (see (3)) and where*

$$\beta_n = 2d \log\left(bdn^2\sqrt{\log(da\pi^2 n^2/3\delta)}/6\delta\right) + 4\log(\pi n). \tag{10}$$

*Proof.* For the sake of consistency with the literature, we will reuse, whenever appropriate, the notations in Bogunovic et al. (2016).

Let us set a discretization $\mathcal{S}_n$ of the spatial domain $\mathcal{S} \subseteq [0, 1]^d$. $\mathcal{S}_n$ is of size $\tau_n^d$ and satisfies

$$||\boldsymbol{x} - [\boldsymbol{x}]_n||_1 \le \frac{d}{\tau_n}, \ \forall \boldsymbol{x} \in \mathcal{S}, \tag{11}$$

where $[\boldsymbol{x}]_n = \arg\min_{\boldsymbol{s} \in \mathcal{S}_n} ||\boldsymbol{s} - \boldsymbol{x}||_1$ is the closest point in $\mathcal{S}_n$ to $\boldsymbol{x}$. Note that a uniform grid on $\mathcal{S}$ ensures (11).

Let us now fix $\delta > 0$ and condition on a high-probability event. If $\beta_n = 2\log\frac{\tau_n^d \pi^2 n^2}{3\delta}$, then

$$|f(\boldsymbol{x}, t_n) - \mu_{\mathcal{D}}(\boldsymbol{x}, t_n)| \le \beta_n^{1/2}\sigma_{\mathcal{D}}(\boldsymbol{x}, t_n), \ \forall n \in \mathbb{N}, \forall \boldsymbol{x} \in \mathcal{S}_n \tag{12}$$

with probability at least $1 - \frac{\delta}{2}$. This directly comes from $f(\boldsymbol{x}, t_n) \sim \mathcal{N}\left(\mu_{\mathcal{D}}(\boldsymbol{x}, t_n), \sigma_{\mathcal{D}}^2(\boldsymbol{x}, t_n)\right)$ and from the Chernoff concentration inequality applied to Gaussian tails $\mathbb{P}\left(|f(\boldsymbol{x}, t_n) - \mu_{\mathcal{D}}(\boldsymbol{x}, t_n)| \le \sqrt{\beta_n}\sigma_{\mathcal{D}}(\boldsymbol{x}, t_n)\right) \ge 1 - e^{-\beta_n/2}$. Therefore, our choice of $\beta_n$ ensures that for a given $n$ and a given $\boldsymbol{x} \in \mathcal{S}_n$, $|f(\boldsymbol{x}, t_n) - \mu_{\mathcal{D}}(\boldsymbol{x}, t_n)| \le \beta_n^{1/2}\sigma_{\mathcal{D}}(\boldsymbol{x}, t_n)$ occurs with a probability at least $1 - \frac{3\delta}{\pi^2 n^2 \tau_n^d}$. The union bound taken over $n \in \mathbb{N}$ and $\boldsymbol{x} \in \mathcal{S}_n$ establishes (12) with probability $1 - \frac{\delta}{2}$.

In particular, setting $\tau_n = Ldn^2$ gives that for all $\boldsymbol{x} \in \mathcal{S}$ and all $t_n \in \mathcal{T}$,

$$|f(\boldsymbol{x}, t_n) - f([\boldsymbol{x}]_n, t_n)| \le \frac{1}{n^2} \tag{13}$$

with probability at least $1 - \frac{\delta}{2}$.

Because $\tau_n = L_n dn^2$, $\beta_n$ becomes $2d \log\left(b\sqrt{\log(da\pi^2 n^2/3\delta)}dn^2/6\delta\right) + 4\log(\pi n)$. Using the triangle inequality and combining (12) with (13), we get that with probability $1 - \delta$, $\boldsymbol{x}_n^* = \arg\max_{\boldsymbol{x} \in \mathcal{S}} f(\boldsymbol{x}, t_n)$ satisfies

$$|f(\boldsymbol{x}_n^*, t_n) - \mu_{\mathcal{D}}([\boldsymbol{x}_n^*]_n, t_n)| \le |f([\boldsymbol{x}_n^*]_n, t_n) - \mu_{\mathcal{D}}([\boldsymbol{x}_n^*]_n, t_n)| + |f(\boldsymbol{x}_n^*, t_n) - f([\boldsymbol{x}_n^*]_n, t_n)|$$

$$\le \beta_n^{1/2}\sigma_{\mathcal{D}}([\boldsymbol{x}_n^*]_n, t_n) + \frac{1}{n^2}. \tag{14}$$

We can now upper bound the instantaneous regret of the TVBO algorithm $\mathcal{A}$:

$$r_n = f(\boldsymbol{x}_n^*, t_n) - f(\boldsymbol{x}_n, t_n)$$

$$\le \mu_{\mathcal{D}}([\boldsymbol{x}_n^*]_n, t_n) + \beta_t^{1/2}\sigma_{\mathcal{D}}([\boldsymbol{x}_n^*]_n, t_n) + \frac{1}{n^2} - f(\boldsymbol{x}_n, t_n) \tag{15}$$

$$\le \mu_{\mathcal{D}}(\boldsymbol{x}_n, t_n) + \beta_t^{1/2}\sigma_{\mathcal{D}}(\boldsymbol{x}_n, t_n) + \frac{1}{n^2} - f(\boldsymbol{x}_n, t_n) \tag{16}$$

$$\le 2\beta_t^{1/2}\sigma_{\mathcal{D}}(\boldsymbol{x}_n, t_n) + \frac{1}{n^2}, \tag{17}$$

where (15) follows directly from (14), (16) from the definition of $\boldsymbol{x}_n = \arg\max_{\boldsymbol{x} \in \mathcal{S}} \alpha_{\mathcal{D}}(\boldsymbol{x}, t_n) = \arg\max_{\boldsymbol{x} \in \mathcal{S}} \mu_{\mathcal{D}}(\boldsymbol{x}, t_n) + \beta_t^{1/2}\sigma_{\mathcal{D}}(\boldsymbol{x}, t_n)$ and (17) from (12). □

Lemma A.1 shows that the posterior variance of the surrogate GP (3) plays a key role in the regret bound. Let us now derive a first bound on the cumulative regret $R_T$, by extending an original idea from Srinivas et al. (2012) to the time-varying setting.

**Lemma A.2.** *Let $\{\mathcal{D}_i\}_{i=1,\cdots,T}$ be a sequence of datasets of size $|\mathcal{D}_i| = \min(i, n)$ for $i = 1, \cdots, T$. Let $R_T = \sum_{i=1}^{T} r_i$ be the cumulative regret of a TVBO algorithm with maximal dataset size $n$. Then,*

$$R_T \leq \sqrt{C_1 T \beta_T \left( \gamma_n + \frac{1}{2} \sum_{i=n+1}^{T} \log(1 + \sigma_0^{-2}\sigma_{\mathcal{D}_i}^2(\boldsymbol{x}_i, t_i)) \right)} + \frac{\pi^2}{6} \tag{18}$$

*where*

$$C_1 = \frac{8}{\log(1 + \sigma_0^{-2})}, \tag{19}$$

*and where $\gamma_n$ is the mutual information $\gamma_n = \sum_{i=1}^{n} \log(1 + \sigma_0^{-2}\sigma_{\mathcal{D}_i}^2(\boldsymbol{x}_i, t_i))$.*

*Proof.* We have

$$R_T = \sum_{i=1}^{T} r_i$$

$$\leq 2 \left( \sum_{i=1}^{n} \beta_i^{1/2}\sigma_{\mathcal{D}_i}(\boldsymbol{x}_i, t_i) + \sum_{i=n+1}^{T} \beta_i^{1/2}\sigma_{\mathcal{D}_i}(\boldsymbol{x}_i, t_i) \right) + \frac{\pi^2}{6} \tag{20}$$

$$\leq \sqrt{4T\beta_T \left( \sum_{i=1}^{n} \sigma_{\mathcal{D}_i}^2(\boldsymbol{x}_i, t_i) + \sum_{i=n+1}^{T} \sigma_{\mathcal{D}_i}^2(\boldsymbol{x}_i, t_i) \right)} + \frac{\pi^2}{6} \tag{21}$$

where (20) uses Lemma A.1, the solution to the Basel problem $\sum_{i=1}^{T} i^{-2} \leq \sum_{i=1}^{\infty} i^{-2} = \pi^2/6$ and the fact that the TVBO algorithm has a maximal dataset size $n$ while (21) uses the Cauchy-Schwarz inequality and $\beta_i \leq \beta_T$ for any $i \leq T$.

Going further, we get

$$R_T \leq \sqrt{4\sigma_0^2 T \beta_T \left( \sum_{i=1}^{n} \sigma_0^{-2}\sigma_{\mathcal{D}_i}^2(\boldsymbol{x}_i, t_i) + \sum_{i=n+1}^{T} \sigma_0^{-2}\sigma_{\mathcal{D}_i}^2(\boldsymbol{x}_i, t_i) \right)} + \frac{\pi^2}{6}$$

$$\leq \sqrt{\frac{8}{\log(1 + \sigma_0^{-2})} T \beta_T \left( \frac{1}{2} \sum_{i=1}^{n} \log(1 + \sigma_0^{-2}\sigma_{\mathcal{D}_i}^2(\boldsymbol{x}_i, t_i)) + \frac{1}{2} \sum_{i=n+1}^{T} \log(1 + \sigma_0^{-2}\sigma_{\mathcal{D}_i}^2(\boldsymbol{x}_i, t_i)) \right)} + \frac{\pi^2}{6} \tag{22}$$

$$= \sqrt{C_1 T \beta_T \left( \gamma_n + \frac{1}{2} \sum_{i=n+1}^{T} \log(1 + \sigma_0^{-2}\sigma_{\mathcal{D}_i}^2(\boldsymbol{x}_i, t_i)) \right)} + \frac{\pi^2}{6} \tag{23}$$

where (22) uses the identity $z^2 \leq \sigma_0^{-2} \log(1 + z^2)/\log(1 + \sigma_0^{-2})$ for $z^2 \in [0, \sigma_0^{-2}]$ and where (23) uses the definition of mutual information $\gamma_n$ and $C_1$. □

We now provide an upper bound for the posterior variance $\sigma_{\mathcal{D}}^2(\boldsymbol{x}, t)$.

**Lemma A.3.** *Let $\mathcal{A}$ be a TVBO algorithm, $\mathcal{D}$ be its dataset of observations with maximal size $n$ and $R(n)$ be its response time. Then,*

$$\sigma_{\mathcal{D}}^2(\boldsymbol{x}, t) \leq 1 - C_2 \|\boldsymbol{u}_n\|_2^2, \tag{24}$$

*where $C_2 = \min_{\boldsymbol{x}, \boldsymbol{x}' \in \mathcal{S}} k_S^2(\boldsymbol{x}, \boldsymbol{x}')/(\max_{\omega \in \mathbb{R}} S_T(\omega)/R(0) + \sigma_0^2)$, $S_T$ is the spectral density associated with $k_T$ and where*

$$\boldsymbol{u}_n = (k_T(R(n)), k_T(2R(n)), \cdots, k_T(nR(n))).$$

*Proof.* Let us start by recalling the definition of the posterior variance in (3):

$$\sigma^2_{\mathcal{D}}(\boldsymbol{x},t) = k((\boldsymbol{x},t),(\boldsymbol{x},t)) - \boldsymbol{k}^{\top}((\boldsymbol{x},t),\mathcal{D})\boldsymbol{\Delta}^{-1}\boldsymbol{k}((\boldsymbol{x},t),\mathcal{D})$$

$$= 1 - \boldsymbol{k}^{\top}((\boldsymbol{x},t),\mathcal{D})\boldsymbol{\Delta}^{-1}\boldsymbol{k}((\boldsymbol{x},t),\mathcal{D}) \tag{25}$$

where (25) comes from Assumptions 3.2 and 3.3.

Finding an upper bound on $\sigma^2_{\mathcal{D}}(\boldsymbol{x},t)$ boils down to finding a lower bound on the quadratic form $q_n(\boldsymbol{x},t) = \boldsymbol{k}^{\top}((\boldsymbol{x},t),\mathcal{D})\boldsymbol{\Delta}^{-1}\boldsymbol{k}((\boldsymbol{x},t),\mathcal{D})$. Let us first consider the kernel vector $\boldsymbol{k}((\boldsymbol{x},t),\mathcal{D})$. Because the dataset $\mathcal{D} = \{(\boldsymbol{x}_1,t_1,y_1),\cdots,(\boldsymbol{x}_n,t_n,y_n)\}$ has a fixed maximal dataset size $n$, for every iteration $m > 2n$:

$$\boldsymbol{k}^{\top}((\boldsymbol{x},t),\mathcal{D}) = (k_S(\boldsymbol{x},\boldsymbol{x}_n)k_T(R(n)),\cdots,k_S(\boldsymbol{x},\boldsymbol{x}_1)k_T(nR(n))). \tag{26}$$

The expression in (26) is due to $\mathcal{A}$ having a constant response time $R(n)$ with a dataset size $n$. In such a dataset $\mathcal{D} = \{(\boldsymbol{x}_i,t_i,y_i)\}_{i\in[n]}$, the temporal input $t_i$ is deterministic. In fact, during the first $n$ iterations, $\mathcal{A}$ fills its dataset and has a response time that goes from $R(1)$ to $R(n)$. Next, $\mathcal{A}$ samples at a constant response time $R(n)$ and the first observations are progressively deleted. After $n$ additional iterations, the kernel vector $\boldsymbol{k}^{\top}((\boldsymbol{x},t),\mathcal{D})$ verifies (26).

Then, consider $\boldsymbol{\Delta} = \boldsymbol{k}(\mathcal{D},\mathcal{D}) + \sigma_0^2\boldsymbol{I}$. Because it is positive definite, its spectral decomposition $\boldsymbol{Q}\boldsymbol{\Lambda}\boldsymbol{Q}^{\top}$ exists, where $\boldsymbol{Q} = (\boldsymbol{\phi}_1,\cdots,\boldsymbol{\phi}_n)$ is the orthogonal matrix whose columns are the eigenvectors of $\boldsymbol{\Delta}$, and $\boldsymbol{\Lambda} = \text{diag}\left(\lambda_1 + \sigma_0^2,\cdots,\lambda_n + \sigma_0^2\right)$ is the diagonal matrix comprising the associated eigenvalues.

The quadratic form $q_n(\boldsymbol{x},t)$ becomes

$$q_n(\boldsymbol{x},t) = \boldsymbol{k}^{\top}((\boldsymbol{x},t),\mathcal{D})\boldsymbol{Q}\boldsymbol{\Lambda}^{-1}\boldsymbol{Q}^{\top}\boldsymbol{k}((\boldsymbol{x},t),\mathcal{D})$$

$$= \boldsymbol{v}^{\top}\boldsymbol{\Lambda}^{-1}\boldsymbol{v}$$

$$= \sum_{i=1}^{n}\frac{v_i^2}{\lambda_i + \sigma_0^2}$$

$$\geq \frac{1}{\lambda_1 + \sigma_0^2}\sum_{i=1}^{n}\sum_{j=1}^{n}\sum_{k=1}^{n}k((\boldsymbol{x},t),(\boldsymbol{x}_j,t_j))\phi_{ij}\phi_{ik}k((\boldsymbol{x},t),(\boldsymbol{x}_k,t_k)) \tag{27}$$

where $\boldsymbol{v} = \boldsymbol{Q}^{\top}\boldsymbol{k}((\boldsymbol{x},t),\mathcal{D})$ while (27) is the quadratic form $q_n(\boldsymbol{x},t)$ rewritten with sums instead of matrix products and using the fact that $\lambda_1 \geq \lambda_i$ for any $i \in [n]$. Furthermore, note that under Assumptions 3.2 and 3.3, the largest eigenvalue $\lambda_1$ of the Gram matrix $\boldsymbol{K}$ built from observations collected at frequency $1/R(n)$ can be bounded from above by $\frac{1}{R(n)}\max_{\omega\in\mathbb{R}}S_T(\omega)$, as recently observed by Bardou & Thiran (2025).

Let $\bar{\lambda} = \frac{1}{R(0)}\max_{\omega\in\mathbb{R}}S_T(\omega) > \frac{1}{R(n)}\max_{\omega\in\mathbb{R}}S_T(\omega)$ when $R(n)$ follows Definition 3.5. Then, rearranging the sums, we have

$$q_n(\boldsymbol{x},t) \geq \frac{1}{\bar{\lambda} + \sigma_0^2}\sum_{j=1}^{n}\sum_{k=1}^{n}k((\boldsymbol{x},t),(\boldsymbol{x}_j,t_j))k((\boldsymbol{x},t),(\boldsymbol{x}_k,t_k))\underbrace{\sum_{i=1}^{n}\phi_{ij}\phi_{ik}}_{\delta_{jk}}$$

$$= \frac{1}{\bar{\lambda} + \sigma_0^2}\sum_{i=1}^{n}k^2((\boldsymbol{x},t),(\boldsymbol{x}_i,t_i)) \tag{28}$$

$$= \frac{1}{\bar{\lambda} + \sigma_0^2}\sum_{i=1}^{n}k_S^2(\boldsymbol{x},\boldsymbol{x}_i)k_T^2(t,t_i) \tag{29}$$

where $\delta_{jk}$ is the Kronecker delta with value 1 if $j = k$ and 0 otherwise, (28) is due to the orthonormality of the eigenvectors of $\boldsymbol{K}$ and (29) is due to Assumption 3.2.

Let $\kappa = \min_{\boldsymbol{x},\boldsymbol{x}'\in\mathcal{S}}k_S(\boldsymbol{x},\boldsymbol{x}')$. Then, $\kappa > 0$ as per Assumption 3.3 and we finally have

$$q_n(\boldsymbol{x},t) \geq \frac{\kappa^2}{\bar{\lambda} + \sigma_0^2}\sum_{i=1}^{n}k_T^2(t,t_i)$$

$$= C_2\|\boldsymbol{u}_n\|_2^2,$$

where $C_2 = \kappa^2 / (\bar{\lambda} + \sigma_0^2)$.

Noting that $\sigma_{\mathcal{D}}^2(\boldsymbol{x}, t) = 1 - q_n(\boldsymbol{x}, t) \leq 1 - C_2 ||\boldsymbol{u}_n||_2^2$ concludes the proof. □

Finally, let us prove Theorem 3.6.

*Proof.* Combining Lemmas A.2 and A.3, we get

$$R_T \leq \sqrt{C_1 T \beta_T \left( \gamma_n + \frac{1}{2} \sum_{i=n+1}^{T} \log(1 + \sigma_0^{-2} (1 - C_2 ||\boldsymbol{u}_n||_2^2)) \right) + \frac{\pi^2}{6}}$$

$$\leq \sqrt{C_1 T \beta_T \left( \gamma_n + \frac{\sigma_0^{-2}}{2} (T - n) (1 - C_2 ||\boldsymbol{u}_n||_2^2) \right) + \frac{\pi^2}{6}}, \tag{30}$$

where (30) uses $\log(1 + x) \leq x$ for any $x > 0$. □

For the sake of completeness, we conclude this appendix by showing that the information gain $\gamma_n$ scales linearly with $n$ in the time-varying setting.

**Lemma A.4.** *For all $n \in \mathbb{N}$, $\gamma_n \in \mathcal{O}(n)$.*

*Proof.* Recall that $\gamma_n$ can be recovered from the covariance matrix $k(\mathcal{D}, \mathcal{D})$ and its spectrum. More specifically, denoting $\{\lambda_i\}_{1 \leq i \leq n}$ the ordered spectrum of $k(\mathcal{D}, \mathcal{D})$, we have

$$\gamma_n = \frac{1}{2} \sum_{i=1}^{n} \log(1 + \sigma_0^{-2} \lambda_i)$$

$$\leq \frac{1}{2} \sigma_0^{-2} \sum_{i=1}^{n} \lambda_i \tag{31}$$

$$= \frac{1}{2} \sigma_0^{-2} \operatorname{Tr}(k(\mathcal{D}, \mathcal{D})) \tag{32}$$

$$= \frac{1}{2} \sigma_0^{-2} n, \tag{33}$$

where (31) uses the fact that $\log(1 + x) \leq x$ for all $x \in \mathbb{R}_{>0}$, (32) uses the fact that the trace of a matrix is the sum of its eigenvalues, and (33) holds for any kernel that is stationary and normalized (i.e., that satisfies $k((\boldsymbol{x}, t), (\boldsymbol{x}, t)) = 1$ for all $(\boldsymbol{x}, t) \in \mathcal{S} \times \mathcal{T}$).

One immediately from (33) that $\gamma_n \in \mathcal{O}(n)$. That concludes the proof. □

## B    RECOMMENDED STALE DATA MANAGEMENT POLICY

In this appendix, we prove Theorem 3.8 by connecting the $L^2$-distance between Lipschitz-continuous acquisition functions and the integrated 2-Wasserstein distance between GP posteriors.

*Proof.* Let us start by considering the effect of Assumption 3.7 on the distance at a single point $(\boldsymbol{x}, t) \in \mathcal{S} \times \mathcal{T}$.

$$|\alpha_{\mathcal{D}}(\boldsymbol{x}, t) - \alpha_{\tilde{\mathcal{D}}}(\boldsymbol{x}, t)| \leq L_T || (\mu_{\mathcal{D}}(\boldsymbol{x}, t) - \mu_{\tilde{\mathcal{D}}}(\boldsymbol{x}, t), \sigma_{\mathcal{D}}(\boldsymbol{x}, t) - \sigma_{\tilde{\mathcal{D}}}(\boldsymbol{x}, t)) ||_2$$

$$= L_T \sqrt{(\mu_{\mathcal{D}}(\boldsymbol{x}, t) - \mu_{\tilde{\mathcal{D}}}(\boldsymbol{x}, t))^2 + (\sigma_{\mathcal{D}}(\boldsymbol{x}, t) - \sigma_{\tilde{\mathcal{D}}}(\boldsymbol{x}, t))^2}. \tag{34}$$

Table 3: Noise variance $\sigma_0^2$, cost of single call $R(0)$ in seconds, time horizon $H$ in minutes and approximated temporal lengthscale $l_T$ in minutes for each benchmark. The table also indicates if each benchmark can be viewed as quasi-static ($l_T \gg R(0)$) or if it can be viewed as near-constant ($R(0) \sim R(\lfloor H/R(0) \rfloor)$).

| Benchmark $(d+1)$ | Observ. Noise $\sigma_0^2$ | Cost $R(0)$ (s) | Horizon $H$ (s) | Lengthscale $l_T$ (s) | Quasi-Static | Near-Constant |
|---|---|---|---|---|---|---|
| Shekel (4) | 0.02 | 8.00 | 600 | 174 | No | Yes |
| Hartmann (3) | 0.05 | 8.00 | 600 | 164 | No | Yes |
| Ackley (4) | 0.05 | 0.05 | 600 | 216 | Yes | No |
| Griewank (6) | 0.30 | 0.05 | 600 | 55 | No | No |
| Eggholder (2) | 0.10 | 0.05 | 600 | 22 | No | No |
| Schwefel (4) | 0.25 | 0.05 | 600 | 35 | No | No |
| Hartmann (6) | 0.05 | 0.10 | 600 | 270 | Yes | No |
| Powell (4) | 2.50 | 0.01 | 600 | 552 | Yes | No |
| Temperature (3) | 0.16 | 0.01 | 1800 | 396 | Yes | No |
| WLAN (7) | 1.50 | 0.10 | 600 | 120 | No | No |

We can now bound the squared $L^2$ distance between the two acquisition functions $\alpha_{\mathcal{D}}$ and $\alpha_{\tilde{\mathcal{D}}}$ on any subset $\mathcal{S}' \times \mathcal{T}'$ of the spatio-temporal domain $\mathcal{S} \times \mathcal{T}$:

$$
\|\alpha_{\mathcal{D}} - \alpha_{\tilde{\mathcal{D}}}\|_2^2 = \oint_{\mathcal{S}'} \int_{\mathcal{T}'} \left( \alpha_{\mathcal{D}}(\boldsymbol{x},t) - \alpha_{\tilde{\mathcal{D}}}(\boldsymbol{x},t) \right)^2 d\boldsymbol{x}dt
$$

$$
\leq \oint_{\mathcal{S}'} \int_{\mathcal{T}'} L_T^2 \left( (\mu_{\mathcal{D}}(\boldsymbol{x},t) - \mu_{\tilde{\mathcal{D}}}(\boldsymbol{x},t))^2 + (\sigma_{\mathcal{D}}(\boldsymbol{x},t) - \sigma_{\tilde{\mathcal{D}}}(\boldsymbol{x},t))^2 \right) d\boldsymbol{x}dt \quad (35)
$$

$$
= L_T^2 \oint_{\mathcal{S}'} \int_{\mathcal{T}'} \left( (\mu_{\mathcal{D}}(\boldsymbol{x},t) - \mu_{\tilde{\mathcal{D}}}(\boldsymbol{x},t))^2 + (\sigma_{\mathcal{D}}(\boldsymbol{x},t) - \sigma_{\tilde{\mathcal{D}}}(\boldsymbol{x},t))^2 \right) d\boldsymbol{x}dt
$$

$$
= L_T^2 W_2^2(\mathcal{GP}_{\mathcal{D}}, \mathcal{GP}_{\tilde{\mathcal{D}}}), \quad (36)
$$

where (35) is due to (34) and (36) comes from the definition of the integrated 2-Wasserstein distance on the domain $\mathcal{S}' \times \mathcal{T}'$. Applying a square-root on both sides yields the desired result. $\qquad\square$

## C  NUMERICAL RESULTS

Here, we provide a detailed description of each implemented benchmark and the associated figures. There are two figures associated with each benchmark, showing their average regrets and the size of their datasets throughout the experiment.

In the following, the synthetic benchmarks will be described as functions of a point $\boldsymbol{z}$ in the $d+1$-dimensional spatio-temporal domain $\mathcal{S} \times \mathcal{T}$. More precisely, the point $\boldsymbol{z}$ is explicitly given by $\boldsymbol{z} = (x_1, \cdots, x_d, t)$. Also, we will write $d' = d+1$ for the sake of brevity.

Each benchmark comes with a time horizon $H$. To make the benchmarks noisy, each call to the objective function is perturbed with a centered Gaussian noise of variance $\sigma_0^2$, equal to $1\%$ of the signal variance. Moreover, to control the expensiveness of the benchmarks, each call to the objective function takes $R(0)$ seconds to complete. The values for $H$, $\sigma_0^2$ and $R(0)$ are unknown to the evaluated TVBO algorithms but are provided in Table 3 for the sake of completeness.

**Schwefel.**  The Schwefel function is $d'$-dimensional, and has the form

$$
f(\boldsymbol{z}) = 418.9829 d' - \sum_{i=1}^{d'} z_i \sin\left( \sqrt{|z_i|} \right).
$$

For the numerical evaluation, we set $d' = 4$ and we optimized the function on the domain $[-500, 500]^{d'}$. The results are provided in Figure 4.

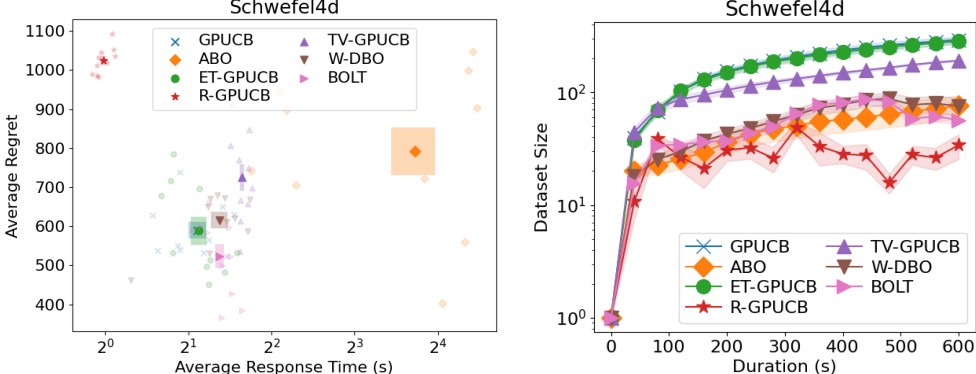

Figure 4: (Left) Average response time and average regrets of the TVBO solutions during the optimization of the Schwefel synthetic function. (Right) Dataset sizes of the TVBO solutions during the optimization of the Schwefel synthetic function.

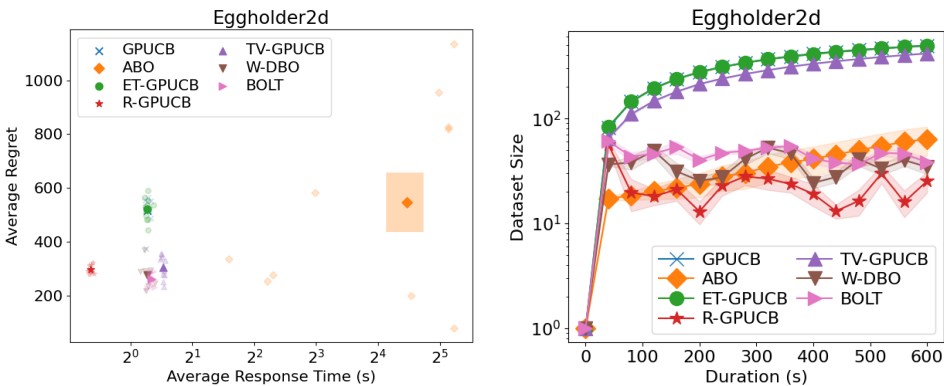

Figure 5: (Left) Average response time and average regrets of the TVBO solutions during the optimization of the Eggholder synthetic function. (Right) Dataset sizes of the TVBO solutions during the optimization of the Eggholder synthetic function.

**Eggholder.** The Eggholder function is 2-dimensional, and has the form

$$f(\boldsymbol{z}) = -(z_2 + 47)\sin\left(\sqrt{\left|z_2 + \frac{z_1}{2} + 47\right|}\right) - z_1 \sin\left(\sqrt{|z_1 - z_2 - 47|}\right).$$

For the numerical evaluation, we optimized the function on the domain $[-512, 512]^2$. The results are provided in Figure 5.

**Ackley.** The Ackley function is $d'$-dimensional, and has the form

$$f(\boldsymbol{z}) = -a\exp\left(-b\sqrt{\frac{1}{d'}\sum_{i=1}^{d'} z_i^2}\right) - \exp\left(\frac{1}{d'}\sum_{i=1}^{d'}\cos(cz_i)\right) + a + \exp(1).$$

For the numerical evaluation, we set $a = 20$, $b = 0.2$, $c = 2\pi$, $d' = 4$ and we optimized the function on the domain $[-32, 32]^{d'}$. The results are provided in Figure 6.

**Shekel.** The Shekel function is 4-dimensional, and has the form

$$f(\boldsymbol{z}) = -\sum_{i=1}^{m}\left(\sum_{j=1}^{4}(z_j - C_{ji})^2 + \beta_i\right)^{-1}.$$

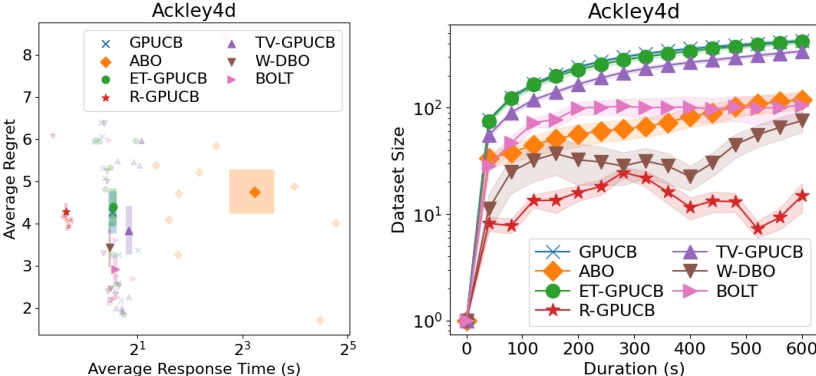

Figure 6: (Left) Average response time and average regrets of the TVBO solutions during the optimization of the Ackley synthetic function. (Right) Dataset sizes of the TVBO solutions during the optimization of the Ackley synthetic function.

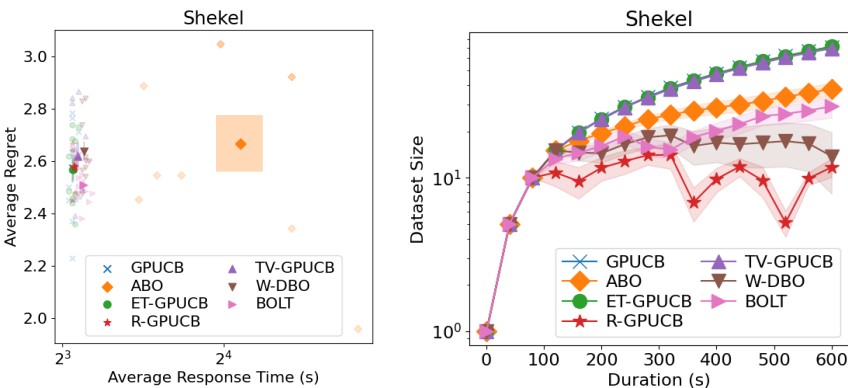

Figure 7: (Left) Average response time and average regrets of the TVBO solutions during the optimization of the Shekel synthetic function. (Right) Dataset sizes of the TVBO solutions during the optimization of the Shekel synthetic function.

For the numerical evaluation, we set $m = 10$, $\boldsymbol{\beta} = \frac{1}{10}(1, 2, 2, 4, 4, 6, 3, 7, 5, 5)$,

$$
\boldsymbol{C} = \begin{pmatrix} 4 & 1 & 8 & 6 & 3 & 2 & 5 & 8 & 6 & 7 \\ 4 & 1 & 8 & 6 & 7 & 9 & 3 & 1 & 2 & 3.6 \\ 4 & 1 & 8 & 6 & 3 & 2 & 5 & 8 & 6 & 7 \\ 4 & 1 & 8 & 6 & 7 & 9 & 3 & 1 & 2 & 3.6 \end{pmatrix},
$$

and we optimized the function on the domain $[0, 10]^4$. The results are provided in Figure 7.

**Griewank.** The Griewank function is $d'$-dimensional, and has the form

$$
f(\boldsymbol{z}) = \sum_{i=1}^{d'} \frac{x_i^2}{4000} - \prod_{i=1}^{d'} \cos\left(\frac{x_i}{\sqrt{i}}\right) + 1
$$

For the numerical evaluation, we set $d' = 6$ and we optimized the function on the domain $[-600, 600]^{d'}$. The results are provided in Figure 8.

**Hartmann-3.** The Hartmann-3 function is 3-dimensional, and has the form

$$
f(\boldsymbol{z}) = -\sum_{i=1}^{4} \alpha_i \exp\left(-\sum_{j=1}^{3} A_{ij}(z_j - P_{ij})^2\right).
$$

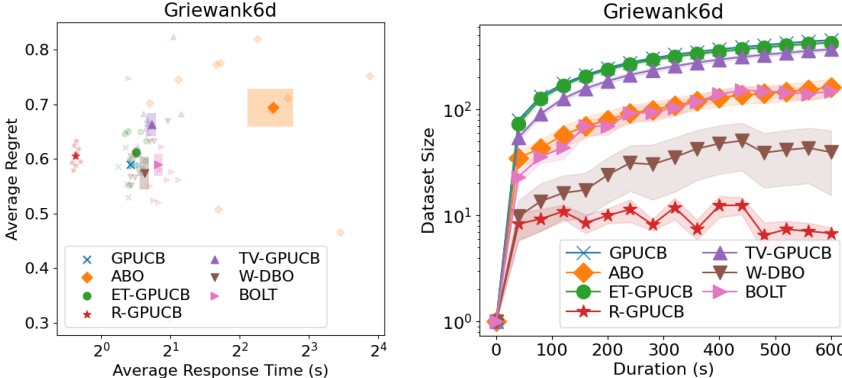

Figure 8: (Left) Average response time and average regrets of the TVBO solutions during the optimization of the Griewank synthetic function. (Right) Dataset sizes of the TVBO solutions during the optimization of the Griewank synthetic function.

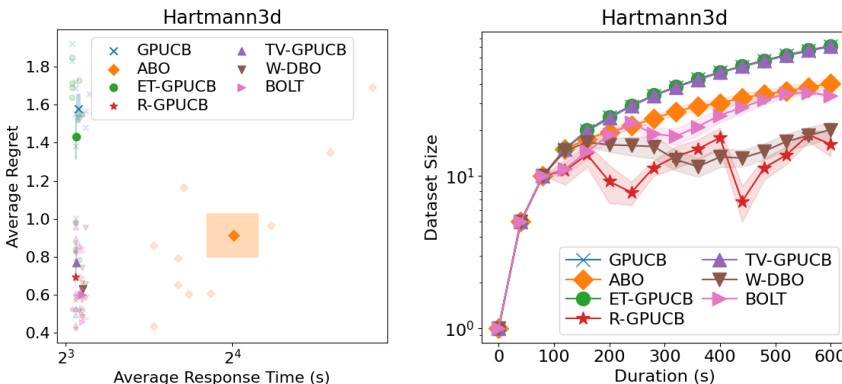

Figure 9: (Left) Average response time and average regrets of the TVBO solutions during the optimization of the Hartmann-3 synthetic function. (Right) Dataset sizes of the TVBO solutions during the optimization of the Hartmann-3 synthetic function.

For the numerical evaluation, we set $\boldsymbol{\alpha} = (1.0, 1.2, 3.0, 3.2)$,

$$
\boldsymbol{A} = \begin{pmatrix} 3 & 10 & 30 \\ 0.1 & 10 & 35 \\ 3 & 10 & 30 \\ 0.1 & 10 & 35 \end{pmatrix}, \boldsymbol{P} = 10^{-4} \begin{pmatrix} 3689 & 1170 & 2673 \\ 4699 & 4387 & 7470 \\ 1091 & 8732 & 5547 \\ 381 & 5743 & 8828 \end{pmatrix},
$$

and we optimized the function on the domain $[0,1]^3$. The results are provided in Figure 9.

**Hartmann-6.** The Hartmann-6 function is 6-dimensional, and has the form

$$
f(\boldsymbol{z}) = -\sum_{i=1}^{4} \alpha_i \exp\left(-\sum_{j=1}^{6} A_{ij}(z_j - P_{ij})^2\right).
$$

For the numerical evaluation, we set $\boldsymbol{\alpha} = (1.0, 1.2, 3.0, 3.2)$,

$$
\boldsymbol{A} = \begin{pmatrix} 10 & 3 & 17 & 3.50 & 1.7 & 8 \\ 0.05 & 10 & 17 & 0.1 & 8 & 14 \\ 3 & 3.5 & 1.7 & 10 & 17 & 8 \\ 17 & 8 & 0.05 & 10 & 0.1 & 14 \end{pmatrix}, \boldsymbol{P} = 10^{-4} \begin{pmatrix} 1312 & 1696 & 5569 & 124 & 8283 & 5886 \\ 2329 & 4135 & 8307 & 3736 & 1004 & 9991 \\ 2348 & 1451 & 3522 & 2883 & 3047 & 6650 \\ 4047 & 8828 & 8732 & 5743 & 1091 & 381 \end{pmatrix},
$$

and we optimized the function on the domain $[0,1]^6$. The results are provided in Figure 10.

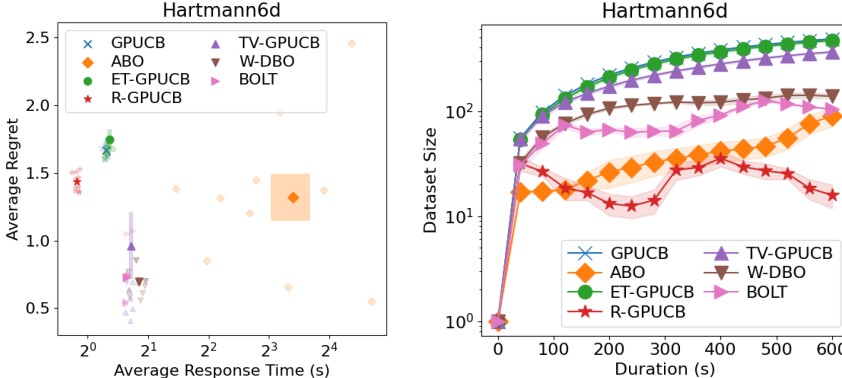

Figure 10: (Left) Average response time and average regrets of the TVBO solutions during the optimization of the Hartmann-6 synthetic function. (Right) Dataset sizes of the TVBO solutions during the optimization of the Hartmann-6 synthetic function.

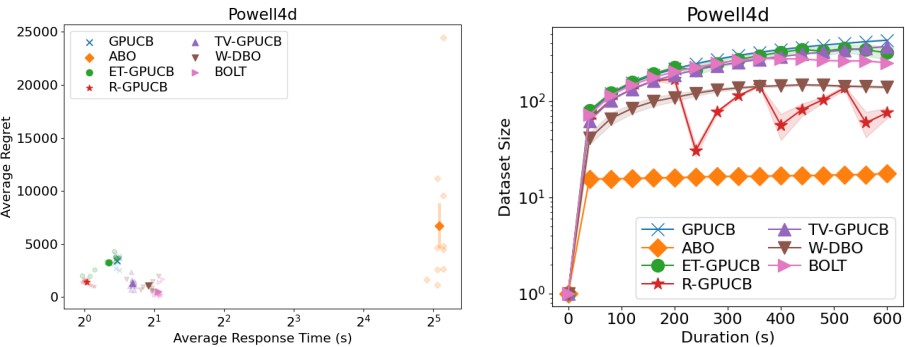

Figure 11: (Left) Average response time and average regrets of the TVBO solutions during the optimization of the Powell synthetic function. (Right) Dataset sizes of the TVBO solutions during the optimization of the Powell synthetic function.

**Powell.** The Powell function is $d'$-dimensional, and has the form

$$f(\boldsymbol{z}) = \sum_{i=1}^{d'/4} (z_{4i-3} + 10z_{4i-2})^2 + 5(z_{4i-1} - z_{4i})^2 + (z_{4i-2} - 2z_{4i-1})^4 + 10(z_{4i-3} - z_{4i})^4.$$

For the numerical evaluation, we set $d' = 4$ and we optimized the function on the domain $[-4, 5]^{d'}$. The results are provided in Figure 11.

**Temperature.** This benchmark comes from the temperature dataset collected from 46 sensors deployed at Intel Research Berkeley. It is a famous benchmark, used in other works such as Bogunovic et al. (2016); Brunzema et al. (2025). The goal of the TVBO task is to activate the sensor with the highest temperature, which will vary with time. To make the benchmark more interesting, we interpolate the data in space-time. With this interpolation, the algorithms can activate any point in space-time, making it a 3-dimensional benchmark (2 spatial dimensions for a location in Intel Research Berkeley, 1 temporal dimension).

For the numerical evaluation, we used the first day of data. The results are provided in Figure 12.

**WLAN.** This benchmark aims at maximizing the throughput of a Wireless Local Area Network (WLAN). 27 moving end-users are associated with one of 6 fixed nodes and continuously stream a large amount of data. As they move in space, they change the radio environment of the network, which should adapt accordingly to improve its performance. To do so, each node has a power level

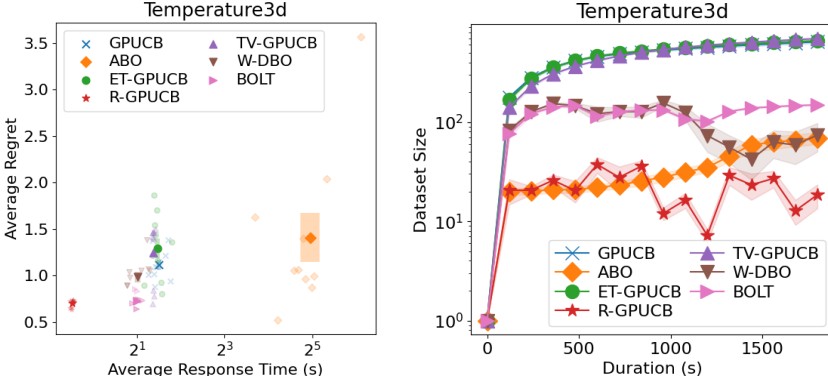

Figure 12: (Left) Average response time and average regrets of the TVBO solutions during the Temperature real-world experiment. (Right) Dataset sizes of the TVBO solutions during the Temperature real-world experiment.

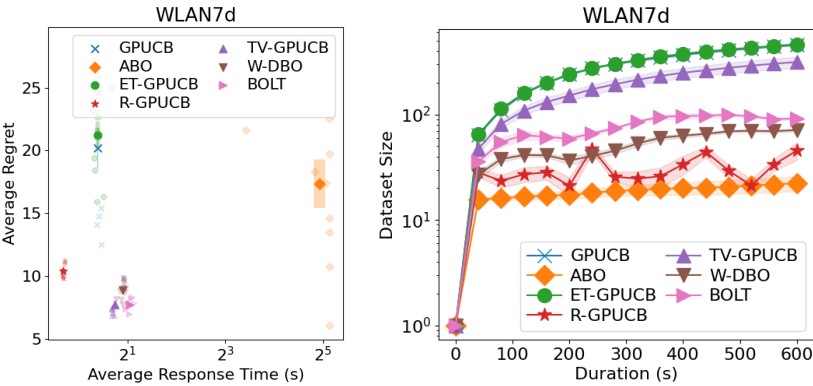

Figure 13: (Left) Average response time and average regrets of the TVBO solutions during the WLAN real-world experiment. (Right) Dataset sizes of the TVBO solutions during the WLAN real-world experiment.

that can be tuned for the purpose of reaching the best trade-off between serving all its users and not causing interference for the neighboring nodes.

The performance of the network is computed as the sum of the Shannon capacities for each pair of node and associated end-users. The Shannon capacity (Kemperman, 1974) sets a theoretical upper bound on the throughput of a wireless communication. We denote it $C(i, j)$, we express it in bits per second (bps). It depends on $S_{ij}$ the Signal-to-Interference plus Noise Ratio (SINR) of the communication between node $i$ and end-user $j$, as well as on $W$, the bandwidth of the radio channel (in Hz):

$$C_{ij}(\boldsymbol{x}, t) = W \log_2(1 + S_{ij}(\boldsymbol{x}, t)).$$

Then, the objective function is

$$f(\boldsymbol{x}, t) = \sum_{i=1}^{6} \sum_{j \in \mathcal{N}_i} C_{ij}(\boldsymbol{x}, t),$$

with $\mathcal{N}_i$ the end-users associated with node $i$.

For the numerical evaluation, we optimized the power levels $\boldsymbol{x}$ in the domain $[10^{0.1}, 10^{2.5}]^6$. For this experiment, the TVBO solutions were evaluated with a Matérn-5/2 for the spatial covariance function and a Matérn-1/2 for the temporal covariance function. The results are provided in Figure 13.

