# OpenReview forum: "Time-Varying Bayesian Optimization Without a Metronome"
_ICLR.cc/2026/Conference — Submitted to ICLR 2026_

### Official Review · Reviewer_yCM5 · 2025-10-27

**Soundness:** 4
**Presentation:** 3
**Contribution:** 4
**Rating:** 8
**Confidence:** 5

**Summary:**

The paper analyses the problems setting of Time-Varying Bayesian Optimisation, where the time intervals between observations are potentially not identical. This is a problem that is likely to occur in real-world applications, as GP is famously slow to fit on large datasets due to its N^3 cost, and if the underlying function changes quickly, waiting too long might mean previously gathered data becomes irrelevant. At the same time, discarding old data too fast will be suboptimal for slowly varying function. The paper attempts to find a solution to this problem. In Theorem 3.6, author derive a novel regret upper bound that explicitly ties the algorithm's regret to the waiting time and pace of change of the time-varying component of the kernel function. Then authors describe, how based on that criterion, the optimal size of dataset can selected in practice. Once deciding on the size of dataset to keep, authors leverage the existing Wasserstein-distance criterion for deciding which points to remove, but provide new insights on it by introducing novel theoretical analysis. Lastly, authors benchmark the proposed algorithm against alternatives.

**Strengths:**

- The problem studied in the paper is very relevant and almost guaranteed to occur in any real-world application of time-varying BO. At the same time, the problem seems to be overlooked in existing literature. As such, this paper fills an important gap in existing research

- The theoretical results authors propose are interesting and insightful. The fact that we can derive practical criterions for selecting dataset sizes to keep based on them makes them very relevant

- The additional insights on the WasserStein-distance point removal heuristic authors provide in Theorem 3.8 nicely complement previous literature

- It looks like the proposed alternative clearly outperforms existing solutions, showing there is a clear practical value in the derived algorithm

- The paper goes beyond merely reporting the regret results and provide a comprehensive analysis of the experiments, showing how the behaviour of their algorithm changes depending on the smoothness of the underlying function

**Weaknesses:**

- The presentation of the paper could be slightly improved. I believe the authors never exactly write what kernel function do they use in their experiments. Such details should be clearly provided, ideally in main body. Also, I believe in the bound in Theorem 3.6, it would be nice to explain why $(1 - C_2||\mathbf{u}_n||_2^2)$ is always greater than zero. Of course, this can be deduced knowing that the temporal kernel decays to zero for large time lags, but to make the paper more accessible, I believe a sentence or two explaining this in the main body would be appropriate.

**Questions:**

- What spatial and temporal kernel functions did you use?
- Can you expand on why solving the optimisation problem in (5) is difficult? It seems to be that you have some objective you need to optimise over natural numbers (up to some maximum horizon length H). As such, you just need H function evaluations, which does not seem like a lot?
- For the relaxed problem you derive, can you quantify the gap between the solution you find and the exact, analytical solution to (5)? Even an empirical comparison would be appreciated.

---

> ### Author Response · Authors · 2025-11-25
>
> We thank the reviewer for the constructive feedback. We address the raised questions and weaknesses below.
>
> > I believe the authors never exactly write what kernel function do they use in their experiments.
>
> Thank you for this suggestion. Actually, we already provide this piece of information in the bullet list of Section 5.1, at the bullet named “Hyperparameters inference”.
>
> > Also, I believe in the bound in Theorem 3.6, it would be nice to explain why $1 - C_2||u_n||_2^2$ is always greater than zero.
>
> Thank you for this suggestion. Upon acceptance, we are planning to add a detailed discussion about the positiveness of $1 - C_2||u_n||_2^2$ to the Appendix A of the camera-ready version.
>
> > Can you expand on why solving the optimisation problem in (5) is difficult?
>
> In Section 3.2, right after Recommendation 1, we state that *finding a closed form for (5) is difficult*. This is because $k_T$ and $R(n)$ are unknown functions that may have widely different expressions in practice. However, as you mentioned, solving this problem using direct search is straightforward, and this is how we do it in practice (see Section 3.2, after Recommendation 1).
>
> > For the relaxed problem you derive, can you quantify the gap between the solution you find and the exact, analytical solution to (5)?
>
> We have not studied the gap between the solution of the problem (5) and its relaxed version. However, note that we are interested in the solution of the integer problem (5). We mention the relaxation of the problem (5) to study its gradient, and show that this relaxed problem has a single maximum. The immediate consequence is that the discretized version of the problem (i.e., the problem (5) we are interested in) also has a single maximum (or two maximums as two consecutive integers). Therefore, our proposed direct search algorithm necessarily converges to the global maximum of (5), that is, $n^*$.
>
> To address the reviewer’s concern, we have added this discussion to Section 3.2.

---

> > ### Comment · Reviewer_yCM5 · 2025-11-26
> >
> > Thank you for your clarifications, I appreciate the additional discussion in Section 3.2, I think it improves the paper.
> >
> > However, there are still some things which are unclear to me. In your response, you write "$k_T$ and $R(n)$ are unknown functions that may have widely different expressions", but you still need to know them to compute the finite different in $u$? Also, what your algorithm essentially does is it starts at some value $n$ and then moves to the next integer that maximises the objective. So, I believe the only advantage over naively checking all $n$ is that if our initial guess is good, we can find it faster (and to the best of my understanding that solution is then the true maximum given your new discussion). As such, I believe the benefit of your approach is potentially faster average case time complexity (with the worst-case time complexity being the same). Thus, I think saying that "finding closed form is difficult" is kind of misleading (it is trivial, just iterate over all n), it would be more appropriate to say it can be time consuming, and the approach you are proposing is speeding it up, which I believe is paramount in the proposed problem setting as you explicitly care about the response time.

---

> > > ### Author Response · Authors · 2025-12-02
> > >
> > > > I think saying that "finding closed form is difficult" is kind of misleading (it is trivial, just iterate over all n).
> > >
> > > In our response, we were referring to the standard definition of a closed-form solution. That is, given an optimization problem $\max_{n \in \mathbb{N}} f(n)$, being able to write $n^* = \arg\max_{n \in \mathbb{N}} f(n) = g(n)$, where $g$ is an explicit formula using a finite number of commonly accepted operations (e.g., addition, product, exponentiation) and functions (e.g., log, tan). Critically, a closed-form solution shall not contain any iterative definition, limit or approximation.
> > >
> > > Therefore, “iterating over all $n$” does not constitute a closed-form solution. In our response, we have agreed with the reviewer that the problem is easy to solve numerically, and that is how we proceed in practice. However, we also wanted to state that finding a closed-form solution that would prevent us from iterating over all $n$ is difficult.
> > >
> > > > In your response, you write "$k_T$ and $R(n)$ are unknown functions that may have widely different expressions", but you still need to know them to compute the finite difference.
> > >
> > > Yes, $k_T$ and $R(n)$ are known at compute time, and we use their expressions to solve the problem numerically. In our response, we wanted to stress that these two functions are problem-dependent. That is precisely the reason why finding a closed-form for $n^*$ is difficult.
> > >
> > > We hope that this additional comment clarifies our initial response.

---

### Official Review · Reviewer_4Wwd · 2025-10-27

**Soundness:** 2
**Presentation:** 2
**Contribution:** 2
**Rating:** 4
**Confidence:** 3

**Summary:**

The paper proposes a method for time-varying Bayesian optimization (TVBO), in which an iteration of BO proceeds with cost R(n) for dataset size n. The regret analysis of the UCB acquisition function is provided for general R(n), which has not been revealed. The proposed algorithm reduces the dataset size into n, specified by the minimization of the regret bound, because the response time diverges if n diverges.

**Strengths:**

The problem setting seems interesting and the theoretical analysis for general R(n) is shown first by the paper according to the authors.

**Weaknesses:**

The problem setting should have been explained in more detail, while the authors mention that it has not been widely studied.

For me, it is currently unclear how theoretical analysis justifies the proposed model. The rate itself of the regret bound is not largely different from existing studies, and the novelty is claimed for revealing the relation between the bound and response time R(n). However, since the bound is quite loose, I do not think the derived bound reveals some essential relation between R(n) and the regret. The data selection algorithm is a kind of simple greedy-like strategy (discard samples having a least effect on the model difference wrt Wasserstein dist at every iteration), and this strategy itself is not fully justified.

**Questions:**

I don't fully understand problem setting (experimental setting). According to appendix, horizon H is 600 sec. Does this mean that BO continues until the sum of the response time becomes 600? The definition of the response time itself is a bit vague. What is 'the time that separates two consecutive iterations'? Usually, BO usually assumes high observation cost for objective function. During the time for waiting observation, 't' proceed?  Does the response time and/or horizon H contain this observation cost? If it contains, what is the observation cost in the experiments? If it does not contain, why?

It is well-known that the standard regret bound by Scrinivas et al 2012 is quite loose, because of which the regret bound (4) should also be quite loose. Therefore, I am not fully sure if the relation between the bound and R(n) has a substantial meaning. Further, in my current understanding, terms removed during deriving the bound also may depend on R(n) (eg., \Delta seemingly depends on t_n), which also makes a substantial meaning of analyzing the relation between the regret bound and R(n) through (4) unclear. As a result, I also do not fully understand how the minimization of (4) can be justified to select n.

Why minimizing (4) is reduced to the minimization of |u_n|^2? Why does the other 'n' in (4) not have any effect?

How is integrated 2-Wasserstein distance calculated? What is the computational complexity? In Bardou et al. (2024b), some approximation is introduced. The same approximation is used? The bound (7) is seemingly quite loose, and so, a quite loose bound is further approximated. How can it be a justification for using integrated 2-Wasserstein distance?

In the paragraph after (5), is the statement 'Starting from n_0 ... converges to n*' proven?

An observation is discarded by W_2 at every time when an additional observation is obtained. Comparing difference between the reduced dataset and the 'full' dataset, can you provide any justification? I guess Theorem 3.8 only (weakly) justifies one time application of the W2 minimization. When the W_2 based selection is applied multiple times, it does not mean that the resulting dataset corresponds to the minimization of the upper bound (7). Further, even if the upper bound (7) is minimized, it is also a quite loose bound.

Theorem 3.6 does not consider how the size n reduced dataset is created, which is actually selected through W_2, i.e., a data-driven manner. Therefore, in the actual process of the proposed algorithm, for example, n and u_n can be seen as a random quantity. Is the inequality in Theorem 3.6 still hold even for those random n and u_n (and C_2?)?

---

> ### Author Response · Authors · 2025-11-25
> **Rebuttal 1/2**
>
> We thank the reviewer for the constructive feedback. We address the raised questions and weaknesses below.
>
> > According to appendix, horizon H is 600 sec. Does this mean that BO continues until the sum of the response time becomes 600?
>
> Yes, our experimental setting is the following: all TVBO algorithms query the objective function as often as they can during $ H = 600$ seconds (except ABO, which can decide when to query next). No algorithm is aware of this termination horizon. Next, each algorithm is evaluated using two different metrics: (i) the average regret $\frac{1}{n} \sum_{i = 1}^n r_i$, where $n$ is the number of queries made by the algorithm, and (ii) their average response time.
>
> > What is 'the time that separates two consecutive iterations'?
>
> A BO loop iteration consists of (i) updating the model with the collected dataset, (ii) finding out where to sample next, and (iii) acquiring a new observation by observing the objective function at that point. We pick a dataset size $n$ and measure the time it takes for the BO algorithm to run (i), (ii) and (iii). This is what we define as the response time $R(n)$.
>
> > During the time for waiting observation, 't' proceed? Does the response time and/or horizon H contain this observation cost? If it contains, what is the observation cost in the experiments? If it does not contain, why?
>
> Yes, our problem setting involves continuous time, so time is always moving forward, including when the algorithm waits for the objective function to be evaluated. By definition, the response time $R(n)$ contains this observation cost. Actually, $R(0)$ is precisely equal to this observation cost. The observation cost, along with other information about each benchmark, is provided in Table 3 of Appendix C.
>
> > The bound is loose, so I am not fully sure if the relation between the bound and $R(n)$ has a substantial meaning.
>
> Thank you for raising this concern. As with most regret bounds (including, as you noted, the regret bound of GP-UCB), it is likely that the bound stated in Theorem 3.6 is loose. Nonetheless, Theorem 3.6 provides the first connection between the regret of a TVBO algorithm and its response time $R(n)$. Importantly, Theorem 3.6 yields a simple and concrete recommendation for the dataset size of a TVBO algorithm. In practice, we observe that a TVBO algorithm that follows this recommendation (BOLT) achieves a significantly lower average regret than the state of the art of TVBO.
>
> We fully expect that future work will refine the relation between $R(n)$ and the cumulative regret of a TVBO algorithm. However, we believe that our result is a first step in this direction, and therefore is a meaningful and useful contribution to the BO community.
>
> In the updated PDF (Section 3.2), we have added a comment about the potential looseness of the bound.
>
> > \Delta seemingly depends on t_n. As a result, I also do not fully understand how the minimization of (4) can be justified to select n.
>
> The matrix $\Delta = k(\mathcal{D}, \mathcal{D}) + \sigma^2_0 I$ indeed depends on $t_1, \cdots, t_n$, the timestamps at which $n$ observations were collected. Our analysis does not assume that $\Delta$ is independent of $t_n$. However, the key idea is that, under Assumption 3.3 (stationarity of the temporal kernel), both the kernel vector $k((x_{n+1}, t_{n+1}), \mathcal{D})$ and the quadratic form involving $\Delta$ can be expressed as a function of $n$ (see (25) and the lower bound on $q_n$).
>
> The dependence on $R(n)$ and, more generally, on $n$ of some terms in the regret bound, is precisely what allows us to connect (perhaps loosely, as you pointed out) the dataset size of a TVBO algorithm and its cumulative regret.
>
> > Why minimizing (4) is reduced to the minimization of |u_n|^2? Why does the other 'n' in (4) not have any effect?
>
> Thank you for raising this concern. The reduction of the minimization in (4) to **maximizing** $||u_n||_2^2$ holds in the long run, under the infinite time horizon that we consider, and we apologize for not making it more explicit.
>
> Concretely, the dataset size $n$ appears in three distinct ways in the bound: (a) inside the information gain $\gamma_n$, (b) as a multiplicative factor $T-n$ and (c) as another multiplicative factor $1 - C_2 ||u_n||_2^2$. With an unknown time horizon $H$, the optimization process runs over an extended period, during which $T$ grows linearly with time while $n$ is a fixed constant chosen by the BO practitioner. Hence, in the long run, $T \gg n$ hence (b) is of the same order as $T$ and (a) is dominated by the other terms in the bound. Therefore, when $T \gg n$, varying $n$ only affects the multiplicative constant through (c), since the only non-vanishing dependence on the choice of $n$ is (c). Minimizing (c) therefore directly minimizes the regret bound in this regime.
>
> In the revision, we have added a discussion about this point before Recommendation 1.

---

> > ### Author Response · Authors · 2025-11-25
> > **Rebuttal 2/2**
> >
> > > How is integrated 2-Wasserstein distance calculated? How can it be a justification for using integrated 2-Wasserstein distance?
> >
> > **Computation.** Because BOLT builds directly upon the open-source implementation of [1], the same approximation strategy is used in all of our experiments.
> >
> > **Justification for its use.** Theorem 3.8 establishes the first theoretical link between stale-data mismatch and the deviation between the acquisition functions $\alpha_\mathcal{D}$ and $\alpha_{\tilde{\mathcal{D}}}$. It shows that the acquisition-function discrepancy can be upper-bounded by the integrated 2-Wasserstein distance between the corresponding surrogate posteriors. We agree that the bound may be loose, but its value lies in providing a rationale for preferring the Wasserstein metric as a stale-data indicator. Prior work [1] motivated the choice empirically; our result is, to our knowledge, the first to offer theoretical support for that choice. Similarly to Theorem 3.6, we fully expect future works from the BO community to refine these results into a more precise connection between acquisition-function discrepancy and integrated 2-Wasserstein distance. Theorem 3.8 provides the right incentive to pursue this research avenue.
> >
> > **In practice.** Finally, this connection leads to a concrete, actionable stale-data management strategy. Combined with the dataset-size recommendation of Recommendation 1, this policy significantly improves performance in practice, as evidenced by BOLT’s results against state-of-the-art TVBO methods.
> >
> > To address the reviewer’s concern, we have revised the motivation and the computational aspects of integrated 2-Wasserstein distances in Section 3.3.
> >
> > > In the paragraph after (5), is the statement 'Starting from n_0 ... converges to n*' proven?
> >
> > Thank you for raising this concern. We have used part of the extra page allowed by ICLR to provide a short proof of the fact that the relaxed version of (5) has a single maximum when the temporal kernel $k_T$ is nonnegative and monotonically decreasing. This immediately implies that the discretized problem (5) also has a unique maximum (or two consecutive integers as maximums). Finally, it is well-known that the simple direct search algorithm we use will converge to a maximum of (5). Since this maximum is unique and global, this necessarily is $n^*$.
> >
> > > Comparing difference between the reduced dataset and the 'full' dataset, can you provide any justification?
> >
> > The reviewer is right, Theorem 3.8 supports the following policy: given a full dataset $\mathcal{D}$ of size $T$ and a desired maximal dataset size $n < T$, remove the observations in $X^\*$, where $X^* = \arg\min_{X \in 2^\mathcal{D}, |X| = T - n} W_2(\mathcal{GP}\_\mathcal{D}, \mathcal{GP}_{\mathcal{D} \setminus X})$.
> >
> > However, this strategy is intractable in the long run, as discussed in Section 4.2 of [1]. Therefore, following the recommendation in [1], BOLT also adopts the tractable heuristic that greedily removes the most stale observation in the dataset according to the integrated 2-Wasserstein distance.
> >
> > > In the actual process of the proposed algorithm, for example, n and u_n can be seen as a random quantity. Is the inequality in Theorem 3.6 still hold even for those random n and u_n?
> >
> > BOLT follows Recommendations 1 and 2, in that order. At each iteration, it proceeds as follows.
> >
> > First, it determines the ideal dataset size $n^\*$ of a TVBO algorithm given two ingredients: the temporal kernel $k_T$ and the response time $R$, following Recommendation 1 derived from Theorem 3.6. Because the optimization task is in a time-varying setting where time only goes forward, Theorem 3.6 operates under the simplifying assumption that the $n$ last observations are kept in the dataset (see Eq. (25) and the surrounding paragraphs). Under this simplifying assumption, $n$ and $u_n$ are arbitrary, but deterministic. This allows Theorem 3.6 to reveal the trade-off between temporal correlation and response time, and therefore guide the choice of $n^*$.
> >
> > Once BOLT has inferred $n^\*$, it applies Recommendation 2 to refine its dataset construction. Instead of simply keeping the last $n^*$ points, it uses the integrated 2-Wasserstein distance to select a more informative subset of observations.
> >
> > In other words, neither Theorem 3.6 nor Theorem 3.8 aim to describe exactly the regret of BOLT. Their goal is to provide two complementary and actionable insights, one for how large the memory budget should be, and one for how stale data should be managed within that budget. These insights inform the design of TVBO algorithms. BOLT is then presented as a TVBO algorithm that integrates these insights in practice.
> >
> > In the revision, we have added a shortened version of the above discussion to Section 4.
> >
> > **References**
> >
> > [1] Bardou, A., Thiran, P., & Ranieri, G. (2024). This Too Shall Pass: Removing Stale Observations in Dynamic Bayesian Optimization. Advances in Neural Information Processing Systems, 37, 42696-42737.

---

### Official Review · Reviewer_Vbpb · 2025-10-28

**Soundness:** 1
**Presentation:** 2
**Contribution:** 1
**Rating:** 2
**Confidence:** 4

**Summary:**

This paper considers the time-varying Bayesian optimization, also known as non-stationary Gaussian process (GP) bandits or kernelized bandits. For this problem, this paper focuses on the computational time of the GP model and proposes a policy to discard the obtained dataset that is far away from the current time. Furthermore, this paper discusses the theoretical result and provides comparisons with several baseline methods.

**Strengths:**

Overall, this paper is well written, and I can understand that the GP model's computational time may be slow, since it is $O(n^3)$.

**Weaknesses:**

Regarding the problem setup and motivation:
- If we adopt the rank-one update of the GP model, the computational time in each step is $O(t^2)$. Bayesian optimization generally considers the case where the objective function evaluation is more dominant compared with the computational time $O(t^2)$. Are there any specific examples where $O(t^2)$ can be a severe bottleneck?
- If the computation of the GP model can be a bottleneck, we can consider other surrogate models or other optimization approaches. Is there no need to discuss and compare such other frameworks?
- Many studies derive efficient approximation techniques for GPs. Is simply using such approximation techniques, such as sparse GPs, not sufficient?

Regarding the theoretical results:
- There is no definition of the maximum information gain $\gamma\_T$. In the time-varying Bayesian optimization, the definition of $\gamma\_T$ is slightly changed and larger than that of the usual setup since the time step is additionally considered. Therefore, to discuss the order of the regret upper bound explicitly, its definition and the order should be described explicitly in the main paper.
- Theorem 3.6 does not seem to be suggestive since it is inconsistent with the actual algorithm. In Eq. (19), it is assumed that the observations after $n$-th one are discarded. However, in the actual algorithm, the observations before $(t - n^\star)$-th one are discarded. Thus, it is inappropriate to choose $n^*$ based on Theorem 3.6.

Others:
- There is a lack of important related works, such as [1].

[1] Shogo Iwazaki, Shion Takeno, Near-Optimal Algorithm for Non-Stationary Kernelized Bandits, Proceedings of the 28th International Conference on Artificial Intelligence and Statistics, vol. 258, pp. 406-414, PMLR, 2025.

- The authors describe that the existing and proposed results are asymptotic. However, since they hold for any $T$, they are not asymptotic.

**Questions:**

Please answer the above questions.

---

> ### Author Response · Authors · 2025-11-25
> **Rebuttal 1/2**
>
> We thank the reviewer for the constructive feedback. We address the raised questions and weaknesses below.
>
> > If we adopt the rank-one update of the GP model, the computational time in each step is $\mathcal{O}(t^2)$. Are there any specific examples where $\mathcal{O}(t^2)$ can be a severe bottleneck?
>
> Thank you for pointing out that a rank-one Cholesky update reduces GP inference complexity from $\mathcal{O}(n^3)$ to $\mathcal{O}(n^2)$. We have updated the PDF accordingly.
>
> Even with this improvement, regret analyses for TVBO ignore so far the unbounded growth of the inference cost of a GP model when more data are accumulated, regardless of the observation cost of the objective. This may be harmless in a static environment, but it becomes a bottleneck in long-running and continuously evolving optimization tasks.
>
> Concretely, in applications where the objective drifts over time, one cannot allow the response time of the BO algorithm to blow up to infinity without loosing the ability to correlate two consecutive observations. Dynamic power control in WLANs is a representative example: it is an ongoing, real-time optimization problem with no predefined termination horizon. In such settings, the cumulative increase of GP inference cost, scaling as $\mathcal{O}(t^2)$ even under rank-one updates, eventually limits the algorithm’s ability to react promptly to environmental changes.
>
> This illustrates why controlling the growth of inference cost remains important, and why the issue we raise is not merely theoretical but directly relevant for real-world, time-evolving optimization scenarios.
>
> > If the computation of the GP model can be a bottleneck, we can consider other surrogate models or other optimization approaches.
>
> Considering alternative surrogates is indeed a possibility, but we have kept our focus on GP-based BO for two reasons. First, all existing regret analyses in TVBO, including the one we conduct, are derived for GP surrogates. Changing the surrogate model would remove the theoretical foundations on which these guarantees rest. Second, in time-varying environments, sample efficiency is even more critical than in static settings: in addition to the usual cost of observing $f$ (money-wise and/or time-wise), observations become obsolete as the objective drifts, and methods that require large datasets (e.g., neural or ensemble surrogates) may perform poorly in this regime. In contrast, the sample efficiency using GP surrogates is well-known, and comes with the additional advantages of their properly calibrated uncertainty and well-established theoretical properties (e.g., universal approximation theorem for some kernels [1]).
>
> Our contribution deals therefore with the computational growth *within* the GP framework, which remains the standard surrogate in TVBO.
>
> > Many studies derive efficient approximation techniques for GPs. Is simply using such approximation techniques, such as sparse GPs, not sufficient?
>
> Approximate GP methods, such as sparse or inducing-point models, can indeed reduce inference cost, but they do not remove the core issue highlighted in the paper. Their complexity remains at least $\mathcal{O}(m^2 t)$ or $\mathcal{O}(m t)$, where $t$ is the number of observations and $m$ the number of inducing points. In time-varying settings, $m$ must grow or the locations of the inducing points must change dynamically to maintain posterior accuracy. This requires a periodic retraining of the sparse model. As a result, the computational cost still grows unbounded and eventually becomes a bottleneck in long-running optimization tasks. Adapting these models to time-varying environments is an interesting contribution, but this is non-trivial and deserves its own paper.
>
> Moreover, all existing regret analyses in TVBO, including the one we derive in this paper, rely on exact GP posteriors. Sparse approximations introduce a bias that is not covered by these theoretical guarantees, it is currently unclear how to adapt the regret framework to these surrogates.
>
> Upon acceptance, we will add a discussion about sparse models to Section 2 of the camera-ready version to address the reviewer’s concern.

---

> > ### Author Response · Authors · 2025-11-25
> > **Rebuttal 2/2**
> >
> > > To discuss the order of the regret upper bound explicitly, the definition of $\gamma_T$ and its order should be described explicitly in the main paper.
> >
> > Thank you for this suggestion. In the time-varying setting, the information gain $\gamma_n$ is still defined as the mutual information $I(y_{1:n}, f_{1:n})$, where $y_{1:n} = (y_1, \cdots, y_n)$ with $y_i = f(x_i, t_i) + \epsilon$ and $f_{1:n} = (f(x_1, t_1), \cdots, f(x_n, t_n))$. Let $K_n$ be the covariance matrix for the observations $f_{1:n}$ (i.e., $K_n = k(\mathcal{D}, \mathcal{D})$ with $\mathcal{D} = ((x_1, t_1), \cdots, (x_n, t_n))$) and $\\{\lambda_i\\}\_{i = 1, \cdots, n}$ its ordered spectrum. Then, we have $\gamma_n = \frac{1}{2} \sum_{i = 1}^n \log(1 + \sigma^{-2} \lambda_i)$. Observing that $\log(1 + \lambda) \leq \lambda$ for all $\lambda > 0$, we have $\gamma_n \leq \frac{1}{2} \sigma^{-2} \sum_{i = 1}^n \lambda_i = \frac{1}{2} \sigma^{-2} \Tr(K_n)$. Because each $i$-th element of the main diagonal of $K_n$ is given by $k((x_i, t_i), (x_i, t_i)) = 1$ for every stationary normalized kernel $k$, we have $\gamma_n \leq \frac{1}{2} \sigma^{-2} n$.
> >
> > Therefore, even in the time-varying setting,$\gamma_n \in \mathcal{O}(n)$.
> >
> > Following the reviewer’s suggestion, we have added this derivation in Appendix A (see Lemma A.4) and stated explicitly that $\gamma_n$ scales at most linearly with the maximal dataset size $n$.
> >
> > > In Eq. (19), it is assumed that the observations after $n$-th one are discarded.
> >
> > Thank you for pointing out the ambiguity caused by the notation. In our analysis, we consider a TVBO algorithm that maintains a maximum dataset size $n$. The cumulative regret is computed by distinguishing two regimes: (i) the $n$ first iterations during which the dataset grows from size $i = 1, \cdots, n$ (denoted $\mathcal{D}_i$) and (ii) all subsequent iterations, during which the dataset size is fixed and equal to $n$ (denoted $\mathcal{D}_n$).
> >
> > This notation was unfortunately overloaded and could be misread as “the dataset at iteration $n$” rather than “the dataset of size $n$”. We have corrected this in the updated PDF.
> >
> > Importantly, our derivation does not discard the latest observations. This is reflected by Eq. (25): the temporal kernel values of the kernel vector $k((x, t), D)$ for a new observation $(x, t)$ range from $k_T(R(n))$ to $k_T(nR(n))$, exactly matching the retention of the latest observations.
> >
> > We thank the reviewer again for pointing out this notation issue and we have revised the PDF to make this distinction explicit.
> >
> > > There is a lack of important related works.
> >
> > Thank you for pointing out the potential omission of the recent work [2]. This work belongs to another, slightly different line of work (discussed in Section 2.2) that operates under a distinct set of assumptions. First, these works consider discrete time steps, while our analysis is conducted in continuous time and account for the response time $R(n)$ of the TVBO algorithm. Second, and more importantly, these works assume that the variational budget of the objective function is bounded, that is $\sum_{i =1}^T ||f_t - f_{t-1}||_{\infty} \leq V_T$. In practice, this is unrealistic in most settings because this immediately implies that $f$ becomes asymptotically static.
> >
> > Upon acceptance, we will make sure that [2] is added to our bibliography.
> >
> > > Since the results hold for any $T$, they are not asymptotic.
> >
> > Thank you for pointing out this potential source of confusion. Although Theorem 3.6 does hold for any $T$, the recommendation that we derive from it holds only when $T \gg n$ (otherwise, the minimization of the regret bound becomes significantly more complex). That is why we initially wrote about the asymptotic regime.
> >
> > In the updated PDF, we have made it clearer that Theorem 3.6 is not an asymptotic result per se, but rather that Recommendation 1, that follows from Theorem 3.6, holds only when $T \gg n$.
> >
> > **References**
> >
> > [1] Micchelli, C. A., Xu, Y., & Zhang, H. (2006). Universal Kernels. Journal of Machine Learning Research, 7(12).
> >
> > [2] Iwazaki, S., & Takeno, S. (2024). Near-optimal algorithm for non-stationary kernelized bandits. arXiv preprint arXiv:2410.16052.

---

> > > ### Comment · Reviewer_Vbpb · 2025-11-26
> > >
> > > I appreciate the kind clarifications. I have understood my several misunderstandings. However, I still have a concern about the following:
> > >
> > > >In Eq. (19), it is assumed that the observations after $n$-th one are discarded.
> > >
> > > If $\mathcal{D}\_n$ is the last $n$ dataset, is it true that $p(f \mid \mathcal{D}\_n)$ is a GP and Lemma A.1 can be applied?
> > > In the usual setting where $\mathcal{D}\_T$ is given, we can see that
> > > $p(f | \mathcal{D}\_T) \propto p(f) p(x\_1) p(y\_1 \mid f, x\_1) \prod\_{t=2}^T p(y\_t | f, \{(x\_{i}, y\_{i})\}\_{i=1}^{t-1}, x\_t ) p(x\_t | \{(x\_{i}, y\_{i})\}\_{i=1}^{t-1} ) \propto p(f) \prod\_{t=1}^T p(y\_t | f, x\_t ),$
> > > where it is important that $x\_t$ and $f$ are independent given $\{(x\_{i}, y\_{i})\}\_{i=1}^{t-1}$.
> > > By this derivation, we can obtain that $p(f | \mathcal{D}\_T)$ is a GP. (Note that $p(f(x\_t))$ is not Gaussian.)
> > >
> > > On the other hand, if we condition on the last $n$ dataset, the above derivation does not hold.
> > > I believe that a more detailed justification to apply Lemma A.1 is required.
> > >
> > > Note that Bogunovic et al. 2016 avoid this problem by resetting the dataset (not using a sliding window).
> > > Furthermore, Zhou and Shroff 2021 do not employ the frequentist setting, where $f$ is deterministic, and thus, the above problem does not occur.
> > >
> > > ## minor comment
> > >
> > > Regarding the following statement,
> > > >In practice, this is unrealistic in most settings because this immediately implies that $f$  becomes asymptotically static.
> > >
> > > Since $V_T$ can depend on $T$, e.g., $V_T = \Theta (\sqrt{T})$, $f$ is not necessarily static.
> > > Furthermore, since the regret is defined on the discrete time step $t_i$ in this paper, I believe that the use of kernels for discrete or continuous time is not an essential difference.
> > > (I agree that focusing on the response time is a clear difference.)
> > >
> > > ## Regarding the other reviewer's comment: https://openreview.net/forum?id=mZLDfo0Wx6&noteId=npS0bFCuZX
> > >
> > > I agree that the definition of regret seems strange.
> > > It is defined as $R_T = \sum\_{i=1}^T f(x^*\_i, t\_i) - f(x\_i, t\_i)$.
> > > This time step $t\_i$ seems to be defined as, e.g., $t\_{i-1} + R(n)$ when the dataset size is $n$ (from the derivations of Theorem 3.6).
> > > However, I think $t_i$ should be determined beforehand by the application problem's mandatory requirements.
> > > In addition, I still believe that $n$ should be determined by such a reason, e.g., the next $x_t$ must be chosen in some given tolerance time.
> > > Since a regret upper bound of $O(T)$ can always be obtained, this approach does not degrade theoretical performance.

---

> > > > ### Author Response · Authors · 2025-12-02
> > > >
> > > > > Is it true that $f | D_n$ is a GP? What about this derivation? Other authors avoid this problem by resetting the dataset.
> > > >
> > > > We are not sure to understand the reviewer’s point. Similarly to all papers in the Bayesian setting of TVBO, $f$ is a GP by Assumption 3.1. Furthermore, it is known that given any finite set of observations $D_n$, $f | D_n$ is also a GP [1]. In [2], resetting the dataset is not a necessary condition for $f$ to be a GP.
> > > >
> > > > > Assuming $V_T \in \Theta(\sqrt{T})$ does not make the objective function static.
> > > >
> > > > In our previous comment, we mentioned that assuming that $V_T$ is sublinear makes the function *asymptotically* static. This holds because the difference between $||f_T - f_{T-1}||$ is bound to converge to 0 as $T$ tends to infinity, in order to prevent $V_T = \sum_{i = 1}^T ||f_T - f_{T-1}||$ to scale linearly. As an example, for $V_T \in \Theta(\sqrt{T})$, one trivially shows that $||f_{T} - f_{T-1}|| \in \Theta(1/\sqrt{T})$.
> > > >
> > > > Therefore, although the function is not static per se, its temporal variations shrink over time and eventually $f$ eventually does not experience any variations in the temporal dimension asymptotically.
> > > >
> > > > > The definition of the regret is a bit strange, the sequence of sampling times should be decided beforehand given the problem constraints.
> > > >
> > > > We acknowledge the reviewer’s point, just like any other regret formulation, this definition of regret may not be suitable for every problem. However, we argue that some problems do follow the formulation of regret formulated in this paper. For example, in protein design [3], a TVBO algorithm does not incur any regret as long as it has not made any recommendation to the user.
> > > >
> > > > **References**
> > > >
> > > > [1] Seeger, M. (2004). Gaussian processes for machine learning. International journal of neural systems, 14(02), 69-106.
> > > >
> > > > [2] Bogunovic, I., Scarlett, J., & Cevher, V. (2016, May). Time-varying Gaussian process bandit optimization. In Artificial Intelligence and Statistics (pp. 314-323). PMLR.
> > > >
> > > > [3] Khan, A., Cowen-Rivers, A. I., Grosnit, A., Deik, D. G. X., Robert, P. A., Greiff, V., ... & Bou-Ammar, H. (2023). Toward real-world automated antibody design with combinatorial Bayesian optimization. Cell Reports Methods, 3(1).

---

### Official Review · Reviewer_X51z · 2025-10-29

**Soundness:** 3
**Presentation:** 3
**Contribution:** 1
**Rating:** 2
**Confidence:** 4

**Summary:**

The paper investigates Bayesian optimization in time-varying settings and irregular sampling. Specifically, the paper bounds the cumulative regret of the irregularly observed locations and connects the regret to the sampling frequency. Based on these results the paper derives the optimal size of the dataset and proposes an TVBO algorithm that leverages these results.

**Strengths:**

- The proposed algorithm comes with regret guarantees and  convincing empirical evaluations.
- I found the results on optimal dataset sizes especially intriguing.

**Weaknesses:**

- Unfortunately, the biggest weakness of the paper is that the problem formulation is nonsensical in the sense that a trivial algorithm can achieve constant regret.

 The problem lies in the definition of regret that sums only the regret at the observed locations $R_T = \sum r_i$ and letting the algorithm decide on the sample times $t_i$. A trivial algorithm that will always achieve $\Theta(1)$ regret samples once and never again achieves $R_T = r_1$. Clearly, this solution is non-sensical in the context of time-varying optimization.

 A relevant problem formulation in the irregular sampling setting would need to sum the regret occurred at the last chosen $x$ at a fixed interval despite the algorithm not choosing a *new* $x$.

- Table 1 is misleading. Regret bounds require regularity assumptions on the objective. Claiming BOLT places no assumption on $f$ is wrong.

This is a bit of a nitpick, but without placing restrictions on $f$ we also allow adversarial objectives. For example, for the function
$$
  f(x,t)=\begin{cases}
    2^t, & \text{if $x = x_i$}.\\
    0, & \text{otherwise}.
  \end{cases}
$$
there exists no algorithm that can achieve linear regret. Indeed, the paper places many assumptions on $f$ despite stating otherwise in Table 1.
Still, I appreciate that the regret bounds allow for a wider variety of temporal correlations than those in previous work.

**Questions:**

Questions:

- I think an interesting consequence of Theorem 3.6 is the fact that for many kernels decreasing the response time will lead to a lower regret in the fixed data set setting. Can we recover sublinear regret by sampling fast enough and letting $||u_n||^2_2$ approach 1?

Suggestions:
- I would encourage the authors to look at the setting I described in the weakness section. I think the presented results and the proposed algorithm carry over to the setting where "unobserved" evaluations still incur regret. Albeit some modification to the theory are necessary. Unfortunately, the required changes are to substantial for a rebuttal.
- Minor: Add citation keys to Table 1.

Notes:
- GP inference does not scale as $\mathcal{O}(n^3)$ in the streaming setting (line 330). Adding a new point scales as $\mathcal{O}(n^2$). [1]

[1] Osborne, Michael, and Michael Alan Osborne. _Bayesian Gaussian processes for sequential prediction, optimisation and quadrature_. Diss. Oxford University, UK, 2010.

---

> ### Author Response · Authors · 2025-11-25
>
> We thank the reviewer for the constructive feedback. We address the raised questions and weaknesses below.
>
> > The problem formulation is nonsensical in the sense that a trivial algorithm can achieve constant regret by sampling only once.
>
> We respectfully disagree with this assessment. The setting that we study (i.e., a TVBO algorithm samples the objective function as frequently as it can) is identical to the formulation in [1], closely follows the formulation in [2], and Theorem 3.6 provides a valid *upper bound* on the cumulative regret in this framework. It remains meaningful as it governs the *scaling behavior* of the corresponding average regret.
>
> In fact, in our experiments, the performance of each BO algorithm is evaluated using *average regret*, defined as $\frac{1}{n}\sum_{i=1}^n r_i$, where $r_i$ is the regret incurred at sampling time $t_i$. Under this metric, the “trivial” algorithm proposed by the reviewer, i.e., sampling only once, performs poorly: its average regret becomes the regret of its single evaluation, which is typically large, since the first query is unlikely to be near the optimum. Consequently, this algorithm does not achieve competitive performance in our setting.
>
> This observation is not only speculative. One of the benchmarked methods, ABO, can choose to delay evaluations by optimizing over both the decision and time variables. As a result, ABO often produces very few observations (e.g., see Figure 11), and indeed exhibits substantially worse average regret, exactly as the discussion above predicts.
>
> We hope this clarifies why the trivial single-sample strategy does not neither undermine our problem formulation, nor our empirical evaluation.
>
> > Claiming BOLT places no assumption on $f$ is wrong.
>
> The reviewer is right, the column titles of Table 1 may mislead the reader.
>
> Just as every other BO paper, we are placing assumptions on $f$, which are detailed explicitly in Section 3.1. For lack of space, assumptions that are common to all TVBO algorithms were not listed in Table 1.
>
> To answer the reviewer’s concern, we have modified the column titles of Table 1 in the updated PDF.
>
> > Can we recover sublinear regret by sampling fast enough and letting $||u_n||_2^2$ approach 1?
>
> That is a very interesting question. In our problem setting, the answer is definitely no, because of how the response time $R(n)$ is defined. Keeping all collected observations in the dataset would lead $R(n)$ to diverge (see Definition 3.5 (iii)). Consequently, one would not be able to sample fast enough, would loose the ability to correlate two consecutive observations, and $||u_n||_2^2$ would eventually converge to $0$, not 1. However, in a setting that does not account for the response time of $R(n)$, there may exist some classes of problems for which sampling fast enough would ensure a sublinear regret.
>
> In the revision, we mention this question in Section 6, and we are indeed very interested in addressing it (which is likely to require non-trivial proofs) in a future work.
>
> > GP inference does not scale as $\mathcal{O}(n^3)$ in the streaming setting (line 330).
>
> Thank you for mentioning that a rank-one (Cholesky) update of the covariance matrix reduces the complexity of GP inference $\mathcal{O}(n^2)$. We acknowledge this and we have modified the relevant occurrences of $\mathcal{O}(n^3)$ in the updated PDF.
>
> **References**
>
> [1] Bardou, A., Thiran, P., & Ranieri, G. (2024). This Too Shall Pass: Removing Stale Observations in Dynamic Bayesian Optimization. Advances in Neural Information Processing Systems, 37, 42696-42737.
>
> [2] Nyikosa, F. M., Osborne, M. A., & Roberts, S. J. (2018). Bayesian optimization for dynamic problems. arXiv preprint arXiv:1803.03432.

---

> > ### Comment · Reviewer_X51z · 2025-11-27
> >
> > Thank you for your detailed answers. I am looking forward to your future work on high-frequency TVBO.
> >
> > However, for this paper my main concern regarding the problem formulation stands.
> >
> > > [...] a TVBO algorithm samples the objective function as frequently as it can [...]
> >
> > I couldn't find this version of the problem formulation in the paper.
> >
> > > [...] in our experiments, the performance of each BO algorithm is evaluated using average regret [...]
> >
> > The choice of the average regret metric and the reasoning behind it is not discussed anywhere in the paper. The first mention of it is in Section 5.2.
> >
> > In fact, the theoretical results in the paper are for cumulative regret where the problem formulation essentially allows for the algorithm to choose not only $x_i$ but also $n$. Hence, my critique since a trivial algorithm will achieve constant cumulative regret.

---

> > > ### Author Response · Authors · 2025-12-02
> > >
> > > > I couldn't find this version of the problem formulation in the paper.
> > >
> > > In the problem formulation, we have not defined the sequence $\\{t_i\\}_{i \in \mathbb{N}}$ as a parameter of the TVBO algorithm. Furthermore, $R(n)$, the response time of the algorithm, is precisely defined as the time that separates two consecutive iterations of the algorithm (Definition 3.5) and, as mentioned throughout the paper (e.g., Section 4 about the estimation of the response time), it is problem-dependent and out of the control of the TVBO algorithm.
> > >
> > > That being said, we acknowledge the reviewer’s point. The problem setting should have been formulated more clearly to prevent this potential confusion. We have updated the PDF with remarks that explicitly state the nature of $\\{t_i\\}_{i \in \mathbb{N}}$ and $R(n)$ in Sections 2.1, 2.2 and 3.1. Thank you for pointing out this potential confusion.
> > >
> > > > The choice of the average regret metric and the reasoning behind it is not discussed anywhere in the paper. The first mention of it is in Section 5.2. [...] Theoretical results are for cumulative regret.
> > >
> > > Indeed, our theoretical results hold for the cumulative regret. As mentioned in our response, the cumulative regret is as relevant as the average regret, because the average regret is simply $R_T / T$, where $R_T$ is the cumulative regret and $T$ is the number of iterations of the TVBO algorithm. Therefore, a result such as $R_T \in \mathcal{O}(f(T))$ (note that the asymptotics are w.r.t. the number of iterations $T$) can directly be used for asymptotics on the average regret.
> > >
> > > We do not argue that the average regret is better than the cumulative regret as an overall performance metric in general, but in our problem setting we argue that it is a performance metric better at capturing the performance of a TVBO algorithm, precisely because of the existence of trivial algorithms with constant cumulative regret (as the reviewer mentions in their initial rebuttal).
> > >
> > > Finally, we do acknowledge that the definition of average regret could have been introduced earlier in the paper. We have updated the PDF, and average regret is now discussed in Sections 2.2 and 3.2.
> > >
> > > > The problem formulation essentially allows for the algorithm to choose not only $x$ but also $n$. Hence my critique since a trivial algorithm would achieve constant cumulative regret.
> > >
> > > We do understand the reviewer’s criticism, and we hope that the clarifications brought in the previous comments (and the comments above), have now put the reviewer’s concerns to a rest. Indeed, choosing the queries $x$ and the dataset size $n$ does not prevent the TVBO algorithm from querying the objective as often as it can.

---

### Comment · Area_Chair_svQL · 2025-12-03
**Further Questions from AC**

Dear Authors,

Thanks for the reviewer responses so far.  Due to the unfortunate situation around the review process, I'm trying to look closer than usual at papers in my batch (except when clearly not needed).

I found myself having non-minor concerns on the main theorem that weren't raised by the reviewers, and I think it's best to seek your comments on them.

1) (**Possible Incorrectness?**) Is $C_2$ an "absolute constant" that doesn't vary with $n$?  If so, isn't it problematic that $\|u_n\|^2$ can grow unbounded with $n$?  (e.g., if $k_T(\cdot) = 1$ everywhere -- granted you have suggested setting $T=n$ in that case, but even for $n < T$ the theorem is still meant to be valid) Wouldn't that give a negative argument to the square root, which should be impossible?

2) (**Trivial unless $n \approx T$?**) Suppose that $n \le T/2$, or even $n \le 0.99T$ or similar.  Then the regret bound appears to be $O(T \sqrt{\beta_T (1 - C_2\|u_n\|^2)})$, and since $\beta_T \to \infty$ this means being strictly above $T$ (which is trivial) unless $C_2 \|u_n\|^2 \approx 1$.  However, it seems very unlikely that $C_2 \|u_n\|^2 \approx 1$, because the analysis seems quite loose in terms of constant factors -- if the "best possible" value $C_2^{\rm best}$ gave $C_2^{\rm best} \|u_n\|^2 \approx 1$, and your own derived value were suboptimal by a factor of 2 (say), then this would give $C_2 \|u_n\|^2 \approx \frac{1}{2}$ which is insufficient to "save" the super-linear regret issue.

You do not need to rush too much in responding, as the meta-review deadline has been moved much later than usual.

Area Chair

---

### Meta-Review · Area_Chair_svQL · 2025-12-08

**Summary:**

The scores on this paper were very mixed (8/4/2/2), with particular concerns around the problem formulation and its clarity, and confusion around the regret notions and the theory given.

When looking at the main theorem, I found myself having yet another technical concern.  Looking at Eq (4), it seems impossible for $C_2$ to be a constant independent of $n$, because $\|u_n\|^2$ can grow unbounded with increasing $n$ and thus make the argument to the square root negative.  On the other hand, if $C_2$ does depend on $n$ then its dependence is too hidden yet crucial.

Moreover, unless $n$ is very close to $T$ and/or $C_2 \|u_n\|^2$ is close to 1, the regret bound is higher than linear and thus trivial.  I can’t imagine  or $C_2 \|u_n\|^2$ being close to 1 in most cases, as the proof seems to be based on rather loose steps, like bounding all eigenvalues via the first one.

Based on the above, I believe that the theorem may sometimes be incorrect and/or usually be trivial (strictly higher than $T$).  By similar reasoning, I am not convinced that it is theoretically justified to design by maximizing $\|u_n\|^2$ (though the empirical evidence is welcome).

The authors might argue that linear regret is unavoidable, but I believe a meaningful theorem still needs to be non-trivial in some clear and interpretable time-varying scenarios (e.g., previous works do this by studying the case that $\epsilon = o(1)$ as $T \to \infty$).

Overall, with multiple reviewers and myself having multiple confusions around the problem formulation and theory, I am unable to recommend acceptance in the current form.

**Reviewer Concerns:**

Some of the reviewer concerns have probably been addressed, and others not fully so (e.g., confusion around the regret notion) since they were commented on even after the rebuttal.  I have added new concerns above.

**Reviewer Scores:**

I believe that after learning my new concerns above, some reviewers might lower their score.

---

### Decision · Program_Chairs · 2026-01-26

Reject